# Denise: Deep Robust Principal Component Analysis for Positive Semidefinite Matrices

**Calypso Herrera**  *calypso.herrera@math.ethz.ch*
*Department of Mathematics*
*ETH Zürich*

**Florian Krach**  *florian.krach@math.ethz.ch*
*Department of Mathematics*
*ETH Zürich*

**Anastasis Kratsios**  *anastasis.kratsios@unibas.ch*
*Department Mathematics*
*McMaster University*

**Pierre Ruyssen**  *pierrot@google.com*
*Google Brain*
*Google Zürich*

**Josef Teichmann**  *josef.teichmann@math.ethz.ch*
*Department of Mathematics*
*ETH Zürich*

**Reviewed on OpenReview:** *https://openreview.net/forum?id=D45gGvUZp2*

## Abstract

The robust PCA of covariance matrices plays an essential role when isolating key explanatory features. The currently available methods for performing such a low-rank plus sparse decomposition are matrix specific, meaning, those algorithms must re-run for every new matrix. Since these algorithms are computationally expensive, it is preferable to learn and store a function that nearly instantaneously performs this decomposition when evaluated. Therefore, we introduce Denise, a deep learning-based algorithm for robust PCA of covariance matrices, or more generally, of symmetric positive semidefinite matrices, which learns precisely such a function. Theoretical guarantees for Denise are provided. These include a novel universal approximation theorem adapted to our geometric deep learning problem and convergence to an optimal solution to the learning problem. Our experiments show that Denise matches state-of-the-art performance in terms of decomposition quality, while being approximately 2000× faster than the state-of-the-art, principal component pursuit (PCP), and 200× faster than the current speed-optimized method, fast PCP.

**Keywords:** Low-rank plus sparse decomposition, positive semidefinite matrices, deep neural networks, geometric deep learning, universal approximation

## 1 Introduction

*Robust principal component analysis* (RPCA) aims to find a low rank subspace that best approximates a data matrix $M$ which has corrupted entries. It is defined as the problem of decomposing a given matrix $M$ into the sum of a low rank matrix $L$, whose column subspace gives the principal components, and a sparse matrix S, which corresponds to the outliers' matrix. The standard method via *convex optimization* has significantly worse computation time than the *singular value decomposition* (SVD) (Wright et al., 2009; Xu et al., 2010;

Candes et al., 2011; Chandrasekaran et al., 2010; Hsu et al., 2011; Lin et al., 2011). Recent results developing efficient algorithms for robust PCA contributed to notably reduce the running time (Rodriguez & Wohlberg, 2013; Netrapalli et al., 2014; Chen & Wainwright, 2015; Yi et al., 2016; Cherapanamjeri et al., 2017).

However, in some cases, it is of utmost importance to *instantaneously* produce robust low rank approximations of a given matrix. In particular, in finance we need instantaneously and for long time series of multiple assets, robust low rank estimates of covariance matrices. For instance, this is the case for high-frequency trading (Aït-Sahalia et al., 2010; Aït-Sahalia & Xiu, 2017; 2019). Moreover, it is useful to have *one* procedure applicable to different data that provides such estimates In addition, in applications involving noise, like covariance matrices in finance, it is important to have a procedure that is insensitive to small noise perturbations, which is not the case for classical approaches.

Our contribution lies precisely in this area by introducing a nearly instantaneous algorithm for robust PCA for symmetric positive semidefinite matrices. Specifically, we provide a simple deep learning based algorithm which ensures continuity with respect to the input matrices, such that small perturbations lead to small changes in the output. Moreover, when the deep neural network is trained, only an evaluation of it is needed to decompose any new matrix. Therefore the computation time is negligible, which is an undeniable advantage in comparison with the classical algorithms. To support our claim, theoretical guarantees are provided for (i) the expressiveness of our neural network architecture and (ii) convergence to an optimal solution of the learning problem.

## 2 Related Work

Let $||M||_{\ell_1} = \sum_{i,j} |M_{i,j}|$ denote the $\ell^1$-norm of the matrix $M$. For a given $\lambda > 0$, the RPCA is formulated as
$$\min_{L,S} \text{rank}(L) + \lambda ||S||_{\ell_1} \quad \text{s.t} \quad M = L + S \,.$$

Although it is $\mathcal{NP}$-hard, approximate relaxations of this minimization problem can be solved in polynomial time. The most popular method to solve RPCA is via *convex relaxation* (Wright et al., 2009; Xu et al., 2010; Candes et al., 2011; Chandrasekaran et al., 2010; Hsu et al., 2011; Lin et al., 2011). It consists of a nuclear-norm-regularized matrix approximation which needs a time-consuming full singular value decomposition (SVD) in each iteration. Let $||M||_* = \sum_i |\sigma_i(M)|$ denote the nuclear norm of $M$, i.e. the sum of the singular values of $M$. Then for a given $\lambda > 0$, the problem can be formulated as

$$\min_{L,S} ||L||_* + \lambda ||S||_{\ell_1} \quad \text{s.t} \quad M = L + S. \tag{1}$$

The principal component pursuit (PCP) (Candes et al., 2011) is considered as the state-of-the-art technique and solves (1) by an *alternating directions algorithm* which is a special case of a more general class of augmented Lagrange multiplier (ALM) algorithms known as *alternating directions methods*. The inexact ALM (IALM) (Lin et al., 2011) is an computationally improved version of the ALM algorithm that reduces the number of SVDs needed.

As the previous algorithms need time-consuming SVDs in each iteration, several non convex algorithms have been proposed to solve (1) for a more efficient decomposition of high-dimensional matrices (Rodriguez & Wohlberg, 2013; Netrapalli et al., 2014; Chen & Wainwright, 2015; Yi et al., 2016; Cherapanamjeri et al., 2017). In particular, the fast principal component pursuit (FPCP) (Rodriguez & Wohlberg, 2013) is an alternating minimization algorithm for solving a variation of (1). By incorporating the constraint into the objective, removing the costly nuclear norm term, and imposing a rank constraint on $L$, the problem becomes

$$\min_{L,S} \tfrac{1}{2} ||M - L - S||_F^2 + \lambda ||S||_{\ell_1} \quad \text{s.t.} \quad \text{rank}(L) = r \,.$$

The authors apply an alternating minimization to solve this equation using a partial SVD. The RPCA via gradient descent (RPCA-GD) (Yi et al., 2016) solves (1) via a gradient descent method.

Our work is related to the low rank Cholesky factorization, which, among others, is used to solve semidefinite programs (Burer & Monteiro, 2001; Journée et al., 2008; 2010; Bandeira et al., 2016; De Sa et al., 2014;

Boumal et al., 2016; Li et al., 2019; Ge et al., 2016). We are not only interested in the low rank approximation, but in a *robust* low rank approximation. In that sense, we estimate the low rank approximation of a matrix which can be corrupted by outliers. Therefore, we are using the $\ell_1$ norm instead of the Frobenius norm as it is done in those works.

The closest related works to ours are Song & Shaowei (1997) and (Baes et al., 2019), which both consider neural network approaches to robust PCA. Only the latter of the two provides optimization guarantees; there the authors consider the minimization problem

$$\min_{U \in \mathbb{R}^{n \times k}} \|M - UU^\top\|_{\ell_1}, \tag{2}$$

where a neural network parameterization $U_\theta$ of the matrix $U$ is optimized with gradient descent to find an approximate solution for any fixed $M$. In particular, for every new input $M$ the optimization has to be repeated. In contrast, we train a neural network on a synthetic training dataset such that the learnt parameters can be reused for any unseen matrix $M'$. In particular, our learning objective is much more involved, since we want to find a function that produces good outputs $U_\theta(M)$ for all $M$ of a certain distribution, i.e. a function that generalizes well. While our learning task is more complicated, our method has the advantage of nearly instantaneous evaluation for any new matrix, once the training is finished, compared to (Baes et al., 2019), where a new optimization problem needs to be solved whenever the method is applied to a new matrix $M$.

Other related problems are *matrix factorization* (Lee & Seung, 2001; Ding et al., 2010; Trigeorgis et al., 2014; Kuang et al., 2012), *matrix completion* (Xue et al., 2017; Nguyen et al., 2018; Sedhain et al., 2015), *sparse coding* (Gregor & LeCun, 2010; Ablin et al., 2019), *robust subspace tracking* (He et al., 2011; Narayanamurthy & Vaswani, 2018) and *anomaly detection* (Chalapathy et al., 2017). Solomon et al. (2019) suggested a deep robust PCA algorithm tailored to clutter suppression in ultrasound, which still depends on applying SVDs in each layer of their convolutional recurrent neural network. Our work is similar to Gregor & LeCun (2010) in spirit, since we also train a neural network to perform a complex and otherwise time-consuming task nearly instantaneously. Our method shares many properties with their encoder, including continuity, differentiability, and implicit generalization over the distribution of the training set. While their encoder can only be trained in a supervised manner, relying on classical (and therefore slow) methods to generate labels for the training set, it is possible to use unsupervised training for our method.

A key component of our approach is the universal approximation capability of the deep neural model implementing Denise. This result is not covered by any of the available universal approximation theorems, including those for standard feedforward neural networks (Hornik et al., 1989; Barron, 1992; Kidger & Lyons, 2020) and those concerning non-euclidean geometries (Kratsios & Bilokopytov, 2020). In contrast, our universal approximation result guarantees that we can generically approximate any function encoding both the geometric and algebraic structure of the low-rank plus sparse decomposition problem.

## 3 Denise

We present *Denise*[1], an algorithm that solves the robust PCA for positive semidefinite matrices, using a deep neural network. The main idea is the following: according to the Cholesky decomposition, a positive semidefinite symmetric matrix $L \in \mathbb{R}^{n \times n}$ can be decomposed into $L = UU^\top$. If $U$ has $n$ rows and $r$ columns, then the matrix $L$ will be of rank $r$ or less. In order to obtain the desired decomposition $M = L + S$, we therefore reduce the problem to finding a matrix $U \in \mathbb{R}^{n \times r}$ such that $S := M - UU^\top$ is a sparse matrix, i.e. a matrix that contains a lot of zero entries. In particular, we define the matrix $U = U_\theta(M) \in \mathbb{R}^{n \times r}$ as the output of a neural network. Then the natural objective of the training of the neural network is to achieve sparsity of $S_\theta(M) := M - U_\theta(M)U_\theta(M)^\top$. A good and widely used approximation of this objective is to minimize the $\ell_1$-norm of $S_\theta(M)$ as in (2). To achieve this, the neural network can be trained in a supervised or an unsupervised way, as explained below, depending on the available training dataset. Once Denise is

---

[1]The name *Denise* comes from **De**ep and **Se**midefi**ni**te.

trained, we only need to evaluate it in order to find the low rank plus sparse decomposition

$$M = \underbrace{U_\theta(M)U_\theta(M)^\top}_{L} + \underbrace{M - U_\theta(M)U_\theta(M)^\top}_{S}$$

of any new positive semidefinite matrix $M$. Therefore, Denise considerably outperforms all existing algorithms in terms of speed, as they need to solve an optimization problem for each new matrix.

Moreover, by the construction of $L = U_\theta(M)U_\theta(M)^T$, we can guarantee the positive semidefiniteness of $L$. We note that in practice one may only have access to exogenously manipulated or corrupt (missing data) covariance or correlation matrices which may cause the loss of their positive semidefiniteness. For example, an empirical correlation matrix of stock returns, where the correlation between two stock returns is decreased by the risk manager, may lose its positive semidefiniteness. The issue of non positive semidefiniteness of correlation matrices in option pricing and risk management is well explained in (Rebonato & Jäckel, 1998). We refer to Higham & Strabić (2016) for a detailed discussion of this issue. By contrast, most algorithms do not ensure that $L$ is kept positive semidefinite, which forces them to correct their output at the expense of their accuracy.

### 3.1 Supervised Learning

If a training set is available where for each matrix $M$ an optimal decomposition into $L + S$ is known, then the network can be trained directly to output the correct low rank matrix, by minimizing the supervised loss

$$\Phi_s(\theta) := \mathbb{E}\left[||L - U_\theta(M)U_\theta(M)^\top||_{\ell_1}\right] = \mathbb{E}\left[||S - S_\theta(M)||_{\ell_1}\right] . \tag{3}$$

where the expectation is taken over matrices drawn from an appropriate data-generating distribution from which the training data is sampled. We want the difference $S - S_\theta$ to be as sparse as possible, therefore we use the $\ell_1$-norm, which approximates this objective. Indeed, if this difference is sparse, then also $S_\theta$ is, since the amount of non-zero entries of $S_\theta$ is upper bounded by the sum of those of $S$ and $S - S_\theta$. On the other hand, a small $\ell_2$-norm of $S - S_\theta$ would not imply any upper bound on the non-zero entries of $S_\theta$.

A synthetic dataset of positive semidefinite matrices with known decomposition can be created by simulating Cholesky factors and sparse matrices (Section 5). Moreover, classical methods can be used to generate labels for any set of matrices that a user would like to use as training set, in case the synthetic dataset doesn't encompass the wanted properties. However, this is not necessarily needed, since unsupervised training can be used instead.

### 3.2 Unsupervised Learning

In some applications only the matrix $M$ but no optimal decomposition is known. In this case, the neural network can be trained by minimizing the unsupervised loss function

$$\Phi_u(\theta) := \mathbb{E}\left[||M - U_\theta(M)U_\theta(M)^\top||_{\ell_1}\right] = \mathbb{E}\left[||S_\theta(M)||_{\ell_1}\right] , \tag{4}$$

where, as in the supervised case, the expectation is taken over matrices drawn from an appropriate data-generating distribution from which the training data is sampled.

### 3.3 Combining Supervised Learning and Unsupervised Finetuning

Often the amount of available training data of a real world dataset is limited. Therefore, we consider the following training procedure. First, Denise is trained with the supervised loss function on a large synthetic dataset, where the decomposition is known (Section 5.1). Then the trained network can be finetuned with the unsupervised loss function on a real world training dataset of matrices, where the optimal decomposition is unknown. This way, Denise can incorporate the peculiarities of the real world dataset.

# 4 Theoretical Guarantees for Denise

We provide theoretical guarantees that on every compact subset of symmetric positive semidefinite matrices, the function performing the optimal low-rank plus sparse decomposition can be approximated arbitrarily well by the neural network architecture of Denise. The proofs are presented in Section 6.

## 4.1 Notation

Let $\mathbb{S}_n$ be the set of $n$-by-$n$ symmetric matrices, $P_n \subset \mathbb{S}_n$ the subset of positive semidefinite matrices and $P_{k,n} \subset P_n$ the subset of matrices with rank at most $k \leq n$. We consider a matrix $M = [M_{i,j}]_{i,j} \in P_n$, e.g., a covariance matrix. The matrix $M$ is to be decomposed as a sum of a matrix $L = [L_{i,j}]_{i,j} \in P_{k,n}$ of rank at most $k$ and a sparse matrix $S = [S_{i,j}]_{i,j} \in P_n$. By the Cholesky decomposition (Higham, 2002, Thm 10.9 b), we know that the matrix $L$ can be represented as $L = UU^\top$, where $U = [U_{i,j}]_{i,j} \in \mathbb{R}^{n \times k}$; thus $M = UU^\top + S$.

Let $f_\theta : \mathbb{R}^{n(n+1)/2} \to \mathbb{R}^{nk}$ be a feedforward neural network with parameters $\theta$. As the matrix $M$ is symmetric, the dimension of the input can be reduced from $n^2$ to $n(n+1)/2$ by taking the triangular lower matrix of $M$. Moreover, we convert the triangular lower matrix to a vector. We combine these two transformations in the operator $h$

$$h : \mathbb{S}_n \to \mathbb{R}^{n(n+1)/2}, \quad M \mapsto (M_{1,1}, M_{2,1}, M_{2,2}, \ldots, M_{n,1}, \ldots, M_{n,n})^\top.$$

Similarly, every vector $X$ of dimension $nk$ can be represented as a $n$-by-$k$ matrix with the operator $g$ defined as

$$g : \mathbb{R}^{nk} \to \mathbb{R}^{n \times k}, \quad X \mapsto \begin{pmatrix} X_1 & \cdots & X_k \\ \vdots & & \vdots \\ X_{(n-1)k+1} & \cdots & X_{(n-1)k+k} \end{pmatrix}.$$

Using $h$ and $g$, the matrix $U$ can be expressed as the output of the neural network $U_\theta(M) = g(f_\theta(h(M)))$ and the low rank matrix can be expressed as $L_\theta(M) = \rho(f_\theta(h(M)))$ for

$$\rho : \mathbb{R}^{kn} \to P_{k,n}, \quad X \mapsto g(X)g(X)^\top.$$

We assume to have a set $\mathcal{Z} \subset \mathbb{S}_n \times P_{k,n}$ of training sample matrices $(M, L)$, which is equipped with a probability measure $\mathbb{P}$, i.e. the data-generating distribution. In the supervised case, we assume that $L$ is an optimal low rank matrix for $M$, while in the unsupervised case, where $L$ is not used, it can simply be set to 0. For a given training sample $(M, L)$, the supervised and unsupervised loss functions $\varphi_s, \varphi_u : \Omega \times \mathcal{Z} \to \mathbb{R}$ are defined as

$$\varphi_s(\theta, M, L) = \|L - \rho(f_\theta(h(M)))\|_{\ell_1} \tag{5}$$

and

$$\varphi_u(\theta, M, L) = \|M - \rho(f_\theta(h(M)))\|_{\ell_1}. \tag{6}$$

Then, the overall loss functions as defined in (3) and (4) can be expressed for $\varphi \in \{\varphi_s, \varphi_u\}$

$$\Phi(\theta) = \mathbb{E}_{(M,L) \sim \mathbb{P}} [\varphi(\theta, M, L)].$$

Moreover, the Monte Carlo approximations of these loss functions are given by

$$\hat{\Phi}^N(\theta) = \frac{1}{N} \sum_{i=1}^{N} \varphi(\theta, M_i, L_i), \tag{7}$$

where $(M_i, L_i)$ are i.i.d. samples of $\mathbb{P}$. Denise can be trained using Stochastic Gradient Descent (SGD). A schematic version of these supervised and unsupervised training schemes is given in the pseudo-Algorithm 1.

---

**Algorithm 1** Training of Denise

---

Fix $\theta_0 \in \Omega, N \in \mathbb{N}$
**for** $j \geq 0$ **do**
  Sample i.i.d. matrices $(M_1, L_1), \ldots, (M_N, L_N) \sim \mathbb{P}$
  Compute the gradient $G_j := \frac{1}{N} \sum_{i=1}^{N} \nabla_\theta \varphi(\theta_j, M_i, L_i)$
  Determine a step-size $h_j > 0$
  Set $\theta_{j+1} = \theta_j - h_j G_j$
**end for**

---

### 4.2 Solution Operator to the Learning Problem

Our first result guarantees that there is a (non-linear) solution operator to (2). Thus, there is an optimal low rank plus sparse decomposition for Denise to learn.

**Theorem 4.1.** *Fix a Borel probability measure $\mathbb{P}$ on $P_n$ and set $0 < \varepsilon \leq 1$. Then:*

*(i) For every $M \in P_n$, the set of optimizers, $\underset{U \in \mathbb{R}^{n \times k}}{\mathrm{argmin}} \|M - UU^T\|_{\ell_1}$, is non-empty and every $U \in$*
   $\underset{U \in \mathbb{R}^{n \times k}}{\mathrm{argmin}} \|M - UU^T\|_{\ell_1}$ *satisfies*

$$L := UU^T \in \underset{L \in P_{k,n}}{\mathrm{argmin}} \|M - L\|_{\ell_1}.$$

*(ii) There exists a Borel-measurable function $f : P_n \to \mathbb{R}^{n \times k}$ satisfying for every $M \in P_n$*

$$f(M) \in \underset{U \in \mathbb{R}^{n \times k}}{\mathrm{argmin}} \|M - UU^T\|_{\ell_1}.$$

*(iii) There exists a compact $K_\varepsilon \subseteq P_n$ such that: $\mathbb{P}(K_\varepsilon) \geq 1 - \varepsilon$ and on which $f$ is continuous and we define the function*

$$f^\star : K_\varepsilon \ni M \mapsto f(M)f(M)^\top \in P_{k,n}. \tag{8}$$

Theorem 4.1 (iii) guarantees that the map $f^\star$ is continuous and can be written as the square of a continuous function $f$ from $K_\varepsilon$ to $\mathbb{R}^{n \times k}$.

### 4.3 Novel Universal Approximation Theorem

We introduce a structured subset of $\mathbb{R}^{n \times n}$-valued functions encapsulating the relevant structural properties of the solution map in (8). We fix a compact $X \subset P_n$. Denise's ability to optimally solve (2) is contingent on its ability to uniformly approximate any function in $\sqrt{C}(X, P_{k,n}) :=$ $\left\{ f \in C(X, P_{k,n}) \,\middle|\, \exists \tilde{f} \in C(X, \mathbb{R}^{n \times k}) : f = \tilde{f}\tilde{f}^\top \right\}$.

Unlike $C(X, \mathbb{R}^{n \times k})$, functions in $\sqrt{C}(X, P_{k,n})$ always output meaningful candidate solutions to (2) since they are necessarily low-rank, symmetric, and positive semidefinite matrices. Due to this non-Euclidean structure the next result is not covered by the standard approximation theorems of Hornik et al. (1989) and Kidger & Lyons (2020). Similarly, every function in $\sqrt{C}(X, P_{k,n})$ encodes the algebraic property (8); namely, it admits a point-wise Cholesky-decomposition which is a continuous $\mathbb{R}^{n \times k}$-valued function. Thus, $\sqrt{C}(X, P_{k,n})$ encapsulates more algebraic structure than $C(X, P_{k,n})$ does. This algebraic structure puts approximation in $\sqrt{C}(X, P_{k,n})$ outside the scope of the purely geometric approximation theorems of Kratsios & Bilokopytov (2020).

Our next result concerns the universal approximation capabilities in $\sqrt{C}(X, P_{k,n})$ by the set of all deep neural models $\hat{f} : P_n \to P_{k,n}$ with representation $\hat{f} = \rho \circ f_\theta \circ h$, where $f_\theta : \mathbb{R}^{\frac{n(n+1)}{2}} \to \mathbb{R}^{kn}$ is a deep feedforward network with activation function $\sigma$. Denote the set of all such models by $\mathcal{N}_{\rho,h}^\sigma$.

The width of $\hat{f} \in \mathcal{N}_{\rho,h}^\sigma$ is defined as the width of $f_\theta$. The activation function $\sigma$ defining $f_\theta$ is required to satisfy the following condition of Kidger & Lyons (2020).

**Assumption 4.2.** *The activation function $\sigma \in C(\mathbb{R})$ is non-affine and differentiable at at-least one point with non-zero derivative at that point.*

**Theorem 4.3.** *Let $X \subset P_n$ be compact and let $\sigma \in C(\mathbb{R})$ satisfy Assumption 4.2. For each $\varepsilon > 0$, and each $f \in \sqrt{C}(X, P_{k,n})$, there is an $\hat{f} \in \mathcal{N}_{g,h}^\sigma$ of width at-most $\frac{n(n+2k+1)+4}{2}$ such that:*

$$\max_{M \in X} \left\| f(M) - \hat{f}(M) \right\|_{\ell_1} < \varepsilon. \tag{9}$$

Theorems 4.1 and 4.3 imply that $\mathcal{N}_{\rho,h}^\sigma$ can approximate $f^\star$ with arbitrarily high probability.

**Corollary 4.4.** *Fix a Borel probability measure $\mathbb{P}$ on $P_n$, $0 < \varepsilon \leq 1$, and $\sigma$ satisfying 4.2. Then, there exists some $\hat{f} \in \mathcal{N}_{g,h}^\sigma$ of width at-most $\frac{n(n+2k+1)+4}{2}$ such that*

$$\max_{M \in K_\varepsilon} \left\| f^\star(M) - \hat{f}(M) \right\|_{\ell_1} < \varepsilon, \tag{10}$$

*where $K_\varepsilon$ was defined in Theorem 4.1.*

### 4.4 Convergence of Denise to a Solution Operator of the Supervised Learning Problem

We show that, under the assumption that Denise has identified the optimal weights minimizing the supervised loss function, it converges to an optimal solution $f^\star$ of Theorem 4.1 (iii). This convergence is shown both in terms of the theoretical loss (3) and using its Monte Carlo approximation (7). We therefore operate under the following assumptions.

**Assumption 4.5.** *We assume to have a compact subset $X \subset P_n$ of matrices $M$ such that a continuous function $f : X \to \mathbb{R}^{n \times k}$ satisfying*

$$f(M) \in \underset{U \in \mathbb{R}^{n \times k}}{\arg\min} \|M - UU^T\|_{\ell_1}$$

*for all $M \in X$ exists. Moreover, we assume that for $f^\star(M) := f(M)f(M)^\top$, the training set is given by*

$$\mathcal{Z} := \{(M, L) \mid M \in X, L = f^\star(M)\}$$

*and that we consider a probability measure $\mathbb{P}$ such that $\mathbb{P}(\mathcal{Z}) = 1$.*

By Theorem 4.1, we know that such a set $X$ exists. For any $D \in \mathbb{N}$ let $\mathcal{N}_{\rho,h}^{\sigma,D} \subset \mathcal{N}_{\rho,h}^\sigma$ be the set of neural networks of depth at most $D$ and let $\Theta_D$ be the set of all admissible weights for such neural networks.

**Theorem 4.6.** *Assume Assumption 4.5 holds and let $f^\star$ be as in there. If for every fixed depth $D$, the weights $\theta_D$ of $\hat{f}_{\theta_D} \in \mathcal{N}_{\rho,h}^{\sigma,D}$ are chosen such that $\Phi_s(\theta_D)$ is minimal, then $\|\hat{f}_{\theta_D} - f^\star\|_{\ell_1}$ converges to 0 in mean ($L^1$-norm) as $D$ tends to infinity.*

In the following, we assume the size of the neural network $D$ is fixed and we study the convergence of the Monte Carlo approximation with respect to the number of samples $N$. Moreover, we show that both types of convergence can be combined. To do so, we define $\tilde{\Theta}_D := \{\theta \in \Theta_D \mid |\theta|_2 \leq D\}$, which is a compact subspace of $\Theta_D$. It is straight forward to see that $\Theta_D$ in Theorem 4.6 can be replaced by $\tilde{\Theta}_D$. Indeed, if the needed neural network weights for an $\varepsilon$-approximation have too large norm, then one can increase $D$ until it is sufficiently big.

**Theorem 4.7.** *Assume Assumption 4.5 holds and let $f^\star$ be as in there. For every $D \in \mathbb{N}$, $\mathbb{P}$-a.s.*

$$\hat{\Phi}_s^N \xrightarrow{N \to \infty} \Phi_s \quad \text{uniformly on } \tilde{\Theta}_D.$$

*Let the size of the neural network $D$ be fixed and let $\theta_D$ be as in Theorem 4.6. If for every fixed size $N$ of the training set, the weights $\theta_{D,N} \in \tilde{\Theta}_D$ are chosen such that $\hat{\Phi}_s^N(\theta_{D,N})$ is minimal, then*

$$\Phi_s(\theta_{D,N}) \xrightarrow{N \to \infty} \Phi_s(\theta_D).$$

*In particular, one can define an increasing sequence $(N_D)_{D \in \mathbb{N}}$ in $\mathbb{N}$ such that $\|\hat{f}_{\theta_{D,N}} - f^\star\|_{\ell_1}$ converges to 0 in mean ($L^1$-norm) as $D$ tends to infinity.*

### 4.5 Convergence of Denise in the Unsupervised Learning Problem

Finally, we present the analogous results to Theorems 4.6 and 4.7 in the unsupervised setting. The primary distinction between the supervised and unsupervised settings is that Denise is only guaranteed to converge to a minimum of the unsupervised loss function (6).

**Assumption 4.8.** *We assume to have a compact subset $\tilde{X} \subset P_n$ of matrices $M$ together with a probability measure $\tilde{\mathbb{P}}$ on the set*

$$\tilde{\mathcal{Z}} := \{(M, L) \,|\, M \in \tilde{X}, L = 0\} \tag{11}$$

*that satisfies $\tilde{\mathbb{P}}(\tilde{\mathcal{Z}}) = 1$.*

In the unsupervised learning task we cannot guarantee that Denise converges to any specific target function as we did in Section 4.4. However, we can still show that its output converges to a minimum in terms of the loss function. Therefore, let us define the minimum

$$\Phi_{\min} := \inf_{f \in \sqrt{C}(\tilde{X}, P_{k,n})} \mathbb{E}_{(M,L) \sim \tilde{\mathbb{P}}}[\|M - f(M)\|_{\ell_1}], \tag{12}$$

for which the following result holds.

**Theorem 4.9.** *Under Assumption 4.8, if for every fixed depth $D$, the weights $\theta_D$ of $\hat{f}_{\theta_D} \in \mathcal{N}_{\rho,h}^{\sigma,D}$ are chosen such that $\Phi_u(\theta_D)$ is minimal, then $\Phi_u(\theta_D)$ converges to the minimum $\Phi_{\min}$.*

Similarly, as in Section 4.4, we can also show the convergence of the Monte Carlo approximation in the unsupervised setting.

**Theorem 4.10.** *Under Assumption 4.8, for every $D \in \mathbb{N}$, $\mathbb{P}$-a.s.*

$$\hat{\Phi}_u^N \xrightarrow{N \to \infty} \Phi_u \quad \text{uniformly on } \tilde{\Theta}_D.$$

*Let the size of the neural network $D$ be fixed and let $\theta_D$ be as in Theorem 4.9. If for every fixed size $N$ of the training set, the weights $\theta_{D,N} \in \tilde{\Theta}_D$ are chosen such that $\hat{\Phi}_u^N(\theta_{D,N})$ is minimal, then*

$$\Phi_u(\theta_{D,N}) \xrightarrow{N \to \infty} \Phi_u(\theta_D).$$

*In particular, one can define an increasing sequence $(N_D)_{D \in \mathbb{N}}$ in $\mathbb{N}$ such that $\hat{\Phi}_u^N(\theta_{D,N_D})$ converges to $\Phi_{\min}$ as $D$ tends to infinity.*

## 5 Numerical Results

In this sections we provide numerical results of Denise. We first train Denise with the supervised loss function on a synthetic training dataset and evaluate it on a synthetic test dataset. We also evaluate Denise on a synthetic test dataset which is generated with a different distribution. Finally, we test Denise on a real word dataset before and after finetuning with the unsupervised loss function. The source code is avaible at `https://github.com/DeepRPCA/Denise`.

### 5.1 Supervised Training

We create a synthetic dataset in order to train Denise using the Monte Carlo approximation (7) of the supervised loss function (3). In particular, we construct a collection of $n$-by-$n$ symmetric positive semidefinite matrices $M$ that can be written as

$$M = L_0 + S_0 \tag{13}$$

for a known matrix $L_0$ of rank $k_0 \leq n$ and a known matrix $S_0$ of given sparsity $s_0$. By sparsity we mean the number of zero-valued elements divided by the total number of elements. For example, a sparsity of 0.95 means that 95% of the elements of the matrix are zeros.

To construct a symmetric low rank matrix $L_0$, we first sample $nk_0$ independent standard normal random variables that we arrange into an $n$-by-$k_0$ matrix $U$. Then $L_0$ is defined as $UU^T$.

To construct a symmetric positive semidefinite sparse matrix $S_0$ we first sample a random pair $(i, j)$ with $1 \leq i < j \leq n$ from an uniform distribution. We then construct an $n$-by-$n$ matrix $\tilde{S}_0$ that has only four non-zero coefficients: the off-diagonal elements $(i, j)$ and $(j, i)$ are set to a number $b$ drawn uniformly randomly in $[-1, 1]$, the diagonal elements $(i, i)$ and $(j, j)$ are set to a number $a$ drawn uniformly randomly in $[|b|, 1]$. An example of a $3 \times 3$ matrix with $(i, j) = (1, 2)$, $b = -0.2$ and $a = 0.3$ is the following:

$$\tilde{S}_0 = \begin{pmatrix} 0.3 & -0.2 & 0 \\ -0.2 & 0.3 & 0 \\ 0 & 0 & 0 \end{pmatrix}.$$

This way, the matrix $\tilde{S}_0$ is positive semidefinite. The matrix $S_0$ is obtained by summing different realizations $\tilde{S}_0^{(l)}$, each corresponding to a different pair $(i, j)$, until the desired sparsity is reached.

With this method, we create a synthetic dataset consisting of 10 million matrices for the training set. Other possibilities to generate the training set exist. For example, other distributions or different levels of sparsity can be used. Diversifying the training set can lead to better performance of the trained algorithm.

To implement Denise, we used the machine learning framework Tensorflow (Abadi et al., 2015) with Keras APIs (Chollet et al., 2015). We have tested several neural network architectures, and settled on a simple feed-forward neural network of four layers, with a total of $32 \times n(n+1)/2$ parameters. Moreover, we have tested various sizes, sparsities and ranks for the samples of the training set. All results were similar, hence we only present those using size $n = 20$, sparsity $s_0 = 0.95$ and rank $k_0 = 3$ in the training set. In this setting, we trained our model using 16 Google Cloud TPU-v2 hardware accelerators. Training took around 8 hours (90 epochs), at which point loss improvements were negligible.

### 5.1.1 Evaluation

We create a synthetic test dataset consisting of 10,000 matrices for each of the test settings, using the method presented in Section 5.1. The synthetic dataset introduced in Section 5.1 is composed of randomly generated low rank plus sparse matrices of a certain rank and sparsity. Therefore, a network which performs well on this random test set should also perform well on a real world datasets with the same rank and similar sparsity. The code to generate the synthetic dataset is deterministic by setting a fixed random seed.

We compare Denise against PCP (Candes et al., 2011), IALM (Lin et al., 2011), FPCP (Rodriguez & Wohlberg, 2013) and RPCA-GD (Yi et al., 2016). All algorithms are implemented as part of the LRS matlab library (Sobral et al., 2015; Bouwmans et al., 2016). Evaluation of all the algorithms was done on the same computer[2] for a fair comparison of the inference time.

We compare the rank of the low rank matrix $L$ and the sparsity of the sparse matrix $S$. We determine the *approximated rank* $r(L)$ by the number of eigenvalues of the low-rank $L$ that are larger than $\varepsilon = 0.01$. Similarly, we determine the *approximated sparsity* $s(L)$ by proportion of the entries of the sparse matrix $S$ which are smaller than $\varepsilon = 0.01$ in absolute value.

Moreover, we compare the relative error between the computed low rank matrix $L$ and the low rank matrix $L_0$ (i.e. the low-rank matrix from the synthetic train and test dataset), by rel.error$(L, L_0) = ||L - L_0||_F / ||L_0||_F$. Similarly, we compare the relative error between the computed sparse matrix $S$ and the sparse matrix $S_0$, by rel.error$(S, S_0) = ||S - S_0||_F / ||S_0||_F$.

To enable a fair comparison between the algorithms, we first ensure that the obtained low-rank matrices $L$ all have the same rank. While in FPCP, RPCA-GD and Denise the required rank is set, in PCP and IALM the required rank is depending on the parameter $\lambda$. Therefore, we empirically determined $\lambda$ in order to reach the same rank. In particular, with $\lambda = 0.56/\sqrt{n}$ for the synthetic dataset and $\lambda = 0.64/\sqrt{n}$ for the real dataset, we approximately obtain a rank of 3 for matrices $L$.

In Table 1, we evaluate Denise (trained on the training set with sparsity $s_0 = 0.95$) in 5 test settings with different sparsity $s_0 \in \{0.6, 0.7, 0.8, 0.9, 0.95\}$. Overall Denise obtains comparable results to the state-of-the-art algorithms, while significantly outperforming the other algorithms in terms of inference speed once it is

---

[2]A machine with 2×Intel Xeon CPU E5-2697 v2 (12 Cores) 2.70GHz and 256 GiB of RAM.

trained. This is due to the fact that only one forward pass through the neural network of Denise is needed during evaluation to compute the decomposition. In contrast to this very fast operation, the state-of-the-art algorithms need to solve an iterative optimization algorithm for each new matrix.

Table 1: Comparison between Denise and state of the art algorithms where $L$ is sampled from a standard normal distribution. For different given sparsity $s(S_0)$ of $S_0$, the output properties are the actual rank $r(L)$ of the returned matrix $L$, the sparsity $s(S)$ of the returned matrix $S$ as well as the relative errors rel.error($L$) and rel.error($S$). Additionally we report the training (only applicable for Denise) and inference time. Results are reported as *mean (std)* computed over all samples of the test sets.

| $s(S_0)$ | Algo | $r(L)$ | $s(S)$ | rel.error(L) | rel.error(S) | time train (h) | time inference (ms) |
|---|---|---|---|---|---|---|---|
| | PCP | 2.94 (0.23) | 0.17 (0.02) | 0.51 (0.10) | 2.45 (0.58) | – | 73.52 (21.13) |
| | IALM | 2.92 (0.27) | 0.09 (0.02) | 0.64 (0.09) | 3.10 (0.67) | – | 27.88 (2.45) |
| 0.60 | FPCP | 3.00 (0.00) | 0.02 (0.01) | 0.48 (0.08) | 2.32 (0.61) | – | 16.55 (4.11) |
| | RPCA-GD | 3.00 (0.00) | 0.02 (0.01) | 0.41 (0.17) | 1.97 (0.93) | – | 59.30 (17.52) |
| | Denise | 3.00 (0.00) | 0.02 (0.01) | 0.46 (0.16) | 2.17 (0.74) | 0 | 0.05 (0.00) |
| | PCP | 2.98 (0.13) | 0.19 (0.02) | 0.48 (0.10) | 2.63 (0.67) | – | 92.51 (25.79) |
| | IALM | 2.96 (0.19) | 0.10 (0.02) | 0.63 (0.09) | 3.46 (0.77) | – | 30.57 (2.79) |
| 0.70 | FPCP | 3.00 (0.00) | 0.03 (0.01) | 0.48 (0.08) | 2.63 (0.70) | – | 10.15 (3.77) |
| | RPCA-GD | 3.00 (0.00) | 0.03 (0.02) | 0.39 (0.18) | 2.18 (1.10) | – | 57.45 (17.52) |
| | Denise | 3.00 (0.00) | 0.02 (0.01) | 0.42 (0.15) | 2.26 (0.82) | 0 | 0.05 (0.00) |
| | PCP | 3.00 (0.06) | 0.22 (0.03) | 0.45 (0.10) | 2.93 (0.83) | – | 98.25 (27.72) |
| | IALM | 2.99 (0.11) | 0.11 (0.02) | 0.62 (0.09) | 4.06 (0.95) | – | 29.47 (2.46) |
| 0.80 | FPCP | 3.00 (0.00) | 0.03 (0.02) | 0.47 (0.08) | 3.11 (0.86) | – | 10.11 (4.42) |
| | RPCA-GD | 3.00 (0.00) | 0.04 (0.03) | 0.38 (0.19) | 2.54 (1.39) | – | 48.03 (14.37) |
| | Denise | 3.00 (0.00) | 0.02 (0.01) | 0.37 (0.14) | 2.38 (0.95) | 0 | 0.05 (0.00) |
| | PCP | 3.00 (0.06) | 0.27 (0.05) | 0.41 (0.11) | 3.65 (1.18) | – | 122.84 (29.65) |
| | IALM | 3.00 (0.08) | 0.12 (0.02) | 0.61 (0.10) | 5.39 (1.33) | – | 30.47 (2.86) |
| 0.90 | FPCP | 3.00 (0.00) | 0.04 (0.02) | 0.47 (0.08) | 4.19 (1.20) | – | 16.43 (4.08) |
| | RPCA-GD | 3.00 (0.00) | 0.09 (0.11) | 0.36 (0.21) | 3.26 (2.01) | – | 60.59 (17.04) |
| | Denise | 3.00 (0.00) | 0.03 (0.01) | 0.30 (0.13) | 2.61 (1.24) | 0 | 0.05 (0.00) |
| | PCP | 3.02 (0.13) | 0.30 (0.07) | 0.39 (0.11) | 4.84 (1.75) | – | 124.99 (31.56) |
| | IALM | 3.00 (0.08) | 0.13 (0.03) | 0.60 (0.10) | 7.39 (2.02) | – | 29.67 (2.33) |
| 0.95 | FPCP | 3.00 (0.00) | 0.05 (0.02) | 0.47 (0.08) | 5.77 (1.77) | – | 16.94 (3.58) |
| | RPCA-GD | 3.00 (0.00) | 0.17 (0.24) | 0.34 (0.22) | 4.28 (2.96) | – | 49.67 (13.86) |
| | Denise | 3.00 (0.00) | 0.03 (0.02) | 0.26 (0.13) | 3.12 (1.66) | 8 | 0.05 (0.00) |

### 5.1.2 Evaluation on Differently Generated Synthetic Data

We additionally create 5 synthetic test sets with different sparsity consisting of 10,000 matrices each, using the method presented in Section 5.1 but with a different distribution. In particular, the low-rank matrices are generated using the Student's $t$-distribution (with parameter $k = 5$) instead of using the standard normal distribution. Also in this example, Denise (trained on the original training set with normal distribution and sparsity $s_0 = 0.95$) achieves similar results, while being nearly instantaneous (Table 2).

Table 2 shows that many of the benchmark robust PCA algorithms struggle to produce decompositions with a competitive level of sparsity. This is likely because they are designed for general matrices and can have trouble with the added symmetry and positive definite structure present in this problem, for which Denise has a specialized inductive bias. For example, the symmetric and positive definiteness of the input matrices violate the condition of (Candes et al., 2011, Theorem 1.1) which state that the sparse part $S$ of the input matrix $M$ has uniformly distributed zero entries so that the PCP algorithm can recover the true $L + S$ decomposition of $M$.

Table 2: Comparison between Denise and state of the art algorithms, where $L$ is sampled from a $t$-distribution. For different given sparsity $s(S_0)$ of $S_0$, the output properties are the actual rank $r(L)$ of the returned matrix $L$, the sparsity $s(S)$ of the returned matrix $S$ as well as the relative errors rel.error($L$) and rel.error($S$). Since Denise was not trained in any of the tested settings, we only report the inference time. Results are reported as *mean (std)* computed over all samples of the test set.

| $s(S_0)$ | Algo | $r(L)$ | $s(S)$ | rel.error(L) | rel.error(S) | time (ms) |
|---|---|---|---|---|---|---|
| | PCP | 2.97 (0.18) | 0.18 (0.02) | 0.60 (0.13) | 5.16 (3.27) | 78.81 (22.54) |
| | IALM | 2.95 (0.22) | 0.09 (0.02) | 0.71 (0.10) | 6.04 (3.25) | 30.52 (2.23) |
| 0.60 | FPCP | 3.00 (0.03) | 0.02 (0.01) | 0.48 (0.11) | 3.89 (1.39) | 11.02 (5.02) |
| | RPCA-GD | 3.00 (0.00) | 0.02 (0.01) | 0.51 (0.20) | 4.50 (3.41) | 48.81 (14.70) |
| | Denise | 3.00 (0.00) | 0.01 (0.01) | 0.41 (0.20) | 3.67 (6.97) | 0.05 (0.00) |
| | PCP | 2.99 (0.11) | 0.19 (0.02) | 0.58 (0.13) | 5.73 (3.58) | 88.22 (24.77) |
| | IALM | 2.98 (0.15) | 0.10 (0.02) | 0.70 (0.10) | 6.81 (3.55) | 29.97 (2.28) |
| 0.70 | FPCP | 3.00 (0.03) | 0.02 (0.01) | 0.48 (0.10) | 4.41 (1.53) | 16.68 (4.10) |
| | RPCA-GD | 3.00 (0.00) | 0.02 (0.02) | 0.51 (0.20) | 5.09 (3.77) | 48.47 (14.52) |
| | Denise | 3.00 (0.00) | 0.01 (0.01) | 0.38 (0.20) | 3.91 (8.88) | 0.05 (0.00) |
| | PCP | 3.00 (0.06) | 0.21 (0.03) | 0.56 (0.14) | 6.64 (4.18) | 106.57 (27.99) |
| | IALM | 2.99 (0.11) | 0.10 (0.02) | 0.69 (0.10) | 8.05 (4.13) | 30.95 (2.85) |
| 0.80 | FPCP | 3.00 (0.03) | 0.03 (0.01) | 0.48 (0.11) | 5.28 (1.87) | 10.02 (3.88) |
| | RPCA-GD | 3.00 (0.00) | 0.03 (0.03) | 0.50 (0.21) | 6.00 (4.39) | 57.42 (17.19) |
| | Denise | 3.00 (0.00) | 0.02 (0.01) | 0.35 (0.20) | 4.33 (10.34) | 0.05 (0.00) |
| | PCP | 3.01 (0.10) | 0.24 (0.04) | 0.54 (0.14) | 8.81 (6.37) | 99.10 (26.59) |
| | IALM | 3.00 (0.09) | 0.12 (0.02) | 0.69 (0.10) | 10.89 (6.29) | 17.78 (1.69) |
| 0.90 | FPCP | 3.00 (0.03) | 0.03 (0.02) | 0.47 (0.11) | 7.12 (2.57) | 10.68 (3.27) |
| | RPCA-GD | 3.00 (0.00) | 0.05 (0.07) | 0.49 (0.22) | 8.09 (6.69) | 41.67 (13.21) |
| | Denise | 3.00 (0.00) | 0.02 (0.01) | 0.31 (0.21) | 5.55 (19.75) | 0.05 (0.00) |
| | PCP | 3.03 (0.17) | 0.27 (0.06) | 0.53 (0.14) | 11.83 (8.03) | 105.88 (26.74) |
| | IALM | 3.01 (0.12) | 0.12 (0.03) | 0.69 (0.10) | 14.83 (7.97) | 30.19 (2.18) |
| 0.95 | FPCP | 3.00 (0.02) | 0.03 (0.02) | 0.47 (0.11) | 9.79 (3.77) | 10.27 (3.75) |
| | RPCA-GD | 3.00 (0.00) | 0.08 (0.15) | 0.48 (0.23) | 10.82 (8.53) | 50.14 (14.15) |
| | Denise | 3.00 (0.00) | 0.02 (0.01) | 0.29 (0.20) | 6.85 (17.01) | 0.05 (0.00) |

## 5.2 A Note on the Computation Time of Denise

While applying the trained Denise algorithm is nearly instantaneous, outperforming all competitors, its training is very time intensive. Incorporating the training time into the time measurement (normalized by the test set), i.e. distributing the 8 hours of training time equally to the evaluation of the 10,000 test matrices, yields an evaluation time of 2880 ms per test sample, which is much slower than the other algorithms. Increasing the test set to the same size as the training set with 10 million samples would decrease this training-adjusted measurement time to less than 3 ms, outperforming the competitors again. Since this training-adjusted measurement time is specific to the test set size (which is chosen arbitrarily), we do not show it in the tables. However, this consideration makes it clear that in cases where only view matrices need to be decomposed, Denise does not offer a benefit in terms of computation time over the existing methods if it needs to be trained. On the other hand, if any time improvement at inference is valuable (e.g. in high-frequency trading), Denise offers the possibility to do the computationally heavy part offline by training it for all needed combinations of matrix and output low-rank sizes beforehand such that, at inference, only its fast evaluation time matters. Moreover, if one has to decompose a large number of matrices or if one regularly needs to decompose matrices of a known size (as would be the case in automated video decomposition tasks e.g. for traffic cameras), the usage of Denise can provide a helpful speedup. In particular, we propose to use Denise in cases where its compilation and training can be done offline, leading to a compilation artifact that

can be used to speed up inference significantly similar to amortized inference (Gershman & Goodman, 2014) and inference compilation (Le et al., 2017; Harvey et al., 2019).

## 5.3 Application on S&P500 Stocks Portfolio

We consider a real world dataset of about 1'000 20-by-20 correlation matrices of daily stock returns (on closing prices), for consecutive trading days, shifted every 5 days, between 1989 and 2019. The considered stocks belong to the S&P500 and have been sorted by the GICS sectors[3]. The first 77% of the data is used as training set and the remaining 23% as test set.

We perform a low-rank plus sparse decomposition of these matrices where we choose the low rank to be equal to $k_0 = 3$ as we used it in the synthetic case before. This choice can be made by the user depending on their preference of the number of resulting principle components. Depending on this choice of $k_0$ the supervised training needs to be done on a corresponding synthetic training set with the same rank $k_0$.

Denise, which was trained on the synthetic dataset with $k_0 = 3$, is once evaluated on the real world test set before and once after finetuning it on the (real world) training set (Table 3). The finetuning considerable improves the performance of Denise. Upon inspection we find that Denise offers comparable performance to the leading fastest robust PCA algorithm, namely FPCP, while executing $30\times$ faster. The synthetic test dataset is composed of 10,000 matrices, while here the test dataset contains around 200 matrices. This explains why the computation time of Denise is higher here, as the effort needed to launch the computations is the same no matter whether 10,000 or 200 matrices are evaluated. If repeating the test set such that it has again 10,000 samples, Denise achieves the same speed as on the synthetic dataset (0.05 ms). In particular, Denise has the advantage of becoming (relatively) faster when applied to more samples.

Table 3: Comparison of Denise and Denise with finetuning (FT) to the state of the art algorithms on the S&P500 dataset's test set. We report the finetuning (only applicable for Denise) and inference time. Results are reported as *mean (std)* computed over all samples of the test set.

| Method | $r(L)$ | $s(S)$ | $RE_{ML} = \frac{||M-L||_F}{||M||_F}$ | FT time (s) | inference time (ms) |
|---|---|---|---|---|---|
| PCP | 2.97 (0.54) | 0.33 (0.06) | 0.15 (0.04) | – | 87.09 (0.02) |
| IALM | 2.89 (0.53) | 0.31 (0.06) | 0.15 (0.04) | – | 29.11 (0.00) |
| FPCP | 2.99 (0.13) | 0.24 (0.08) | 0.11 (0.03) | – | 17.91 (0.02) |
| RPCA-GD | 3.00 (0.07) | 0.19 (0.08) | 0.22 (0.05) | – | 61.23 (0.03) |
| Denise | 3.00 (0.00) | 0.08 (0.02) | 0.18 (0.03) | – | 0.66 (0.00) |
| Denise (FT) | 3.00 (0.00) | 0.15 (0.04) | 0.15 (0.04) | 45 | 0.62 (0.00) |

## 5.4 Discussion of the Computational Challenges for Denise

In general, the two main computational challenges in deep learning are high-dimensionality and low-regularity of the target map. While they often appear together, in our experimental setup the dimension of the problem is relatively low for deep learning standards not posing an obstruction. However, the learnability of the highly irregular function performing the robust PCA decomposition truly is a computational challenge shown to be surmounted by Denise in our experiments.

This can be seen, for example, by examining the optimal approximate rates for ReLU neural networks, when approximating continuous functions and smooth functions between Euclidean spaces; see Shen et al. (2022) and Lu et al. (2021), respectively. Consider the case of 400 dimensional inputs, as in the $20 \times 20$ matrices in Section 5.3. The former of these optimal approximation theorems guarantees that the uniform approximation

---

[3]According to the global industry classification standard: energy , materials , industrials, real estate, consumer discretionary, consumer staples, health care, financials, information technology, communication services, utilities.

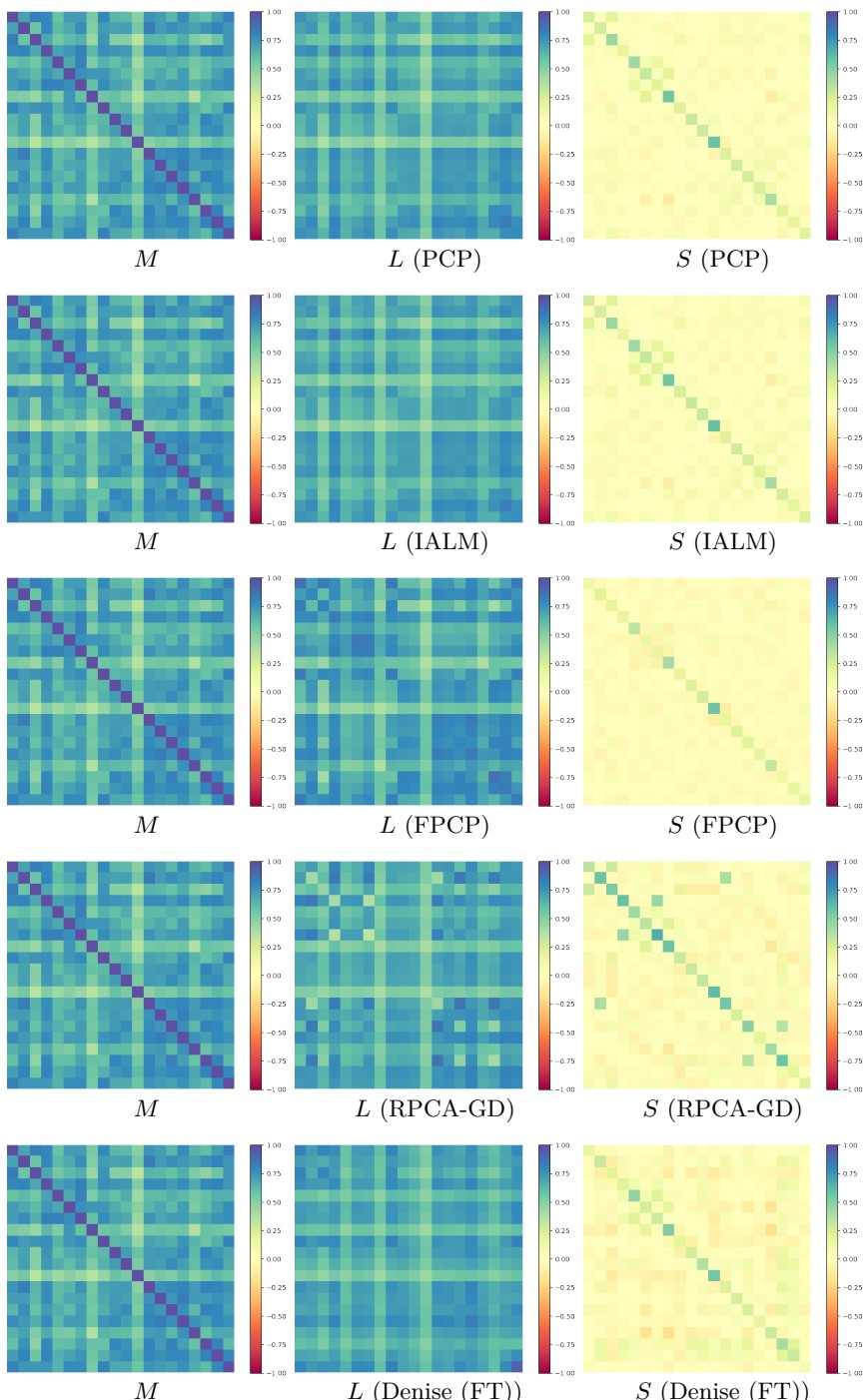

Figure 1: Decomposition into a low-rank plus a sparse matrix of the correlation matrix of a portfolio of 20 stocks among the S&P500 stocks. The forced rank is set to $k = 3$. We have $||M - L||_F/||M||_F$ at 0.15 for PCP, 0.15 for IALM, at 0.11 for FPCP, at 0.22 for RPCA-GD and at 0.15 for Denise. The reconstruction metric $||M - L - S||_F/||M||_F$ is 0 for all algorithms. The computation times in milliseconds are: 103.24 for PCP, 28.66 for IALM, 15.20 for FPCP, 58.17 for RPCA-GD and 0.62 for Denise.

of an $\alpha$-Hölder continuous target function from a compact subset $X \subset P_{20}$ to $P_{k,20}$, to any given precision $\varepsilon > 0$, requires a network depending on roughly $\mathcal{O}(\frac{1}{\varepsilon^{800/\alpha}})$ trainable parameters. The latter one implies that if this target function is sufficiently smooth the number of parameters determining this network can be polynomial in $\frac{1}{\varepsilon}$. The same must be true for Denise, which can be seen by relying on the quantitative non-Euclidean universal approximation theorem of (Kratsios & Papon, 2022, Theorem 9) instead of the qualitative version in Kratsios & Bilokopytov (2020) that we used in the proof of Theorem 4.3.

Theorem 4.1 shows that the function performing the robust PCA decomposition, namely $f^\star$, is highly irregular and thus difficult to learn. This is because it is not smooth on $P_{20}$, but only continuous on a suitable compact set $K_\varepsilon$ thereof. Therefore, our experiments illustrate that Denise can actually learn this highly irregular map, which is provably challenging for any deep learning model. Moreover, it does so while offering competitive performance to any of the state-of-the-art "matrix-wise" algorithms.

# 6 Proofs

## 6.1 Proof of Low Rank Recovery via Universal Approximation

Let $(P_n, dist(A, B) := \|A - B\|_{\ell_1})$ be the metric space of $n \times n$ symmetric positive semidefinite matrices with real coefficient. Let $C(X, P_{k,n})$ be the set of continuous functions from $X$ to $P_{k,n}$, given any (non-empty) subset $X \subset P_n$. Analogously to (Leshno et al., 1993), the set $C(X, P_{k,n})$ is made a topological space, by equipping it with the topology of uniform convergence on compacts, also called compact-convergence, which is generated by the sub-basic open sets of the form

$$B_K(f, \varepsilon) := \left\{ g \in C(X, P_{k,n}) \,\middle|\, \sup_{x \in K} \|f(x) - g(x)\|_{\ell_1} < \varepsilon \right\},$$

where $\varepsilon > 0$, $K \subset X$ compact and $f \in C(X, P_{k,n})$. In this topology, a sequence $\{f_j\}_{j \in \mathbb{N}}$ in $C(X, P_{k,n})$ converges to a function $f \in C(X, P_{k,n})$ if for every non-empty compact subset $K \subseteq X$ and every $\varepsilon > 0$ there exists some $N \in \mathbb{N}$ for which

$$\sup_{x \in K} \|f_j(x) - f(x)\|_{\ell_1} < \varepsilon \qquad \text{for all } j \geq N.$$

This topological space is metrizable. The topology on $\sqrt{C}(X, P_{k,n})$ is the subspace topology induced by inclusion in $C(X, P_{k,n})$ (see (Munkres, 2000, Chapter 18)).

*Proof of Theorem 4.1.* For every $M \in P_n$, the map from $\mathbb{R}^{n \times k}$ to $\mathbb{R}$ defined by $U \to \|M - UU^T\|_{\ell_1}$ is continuous, bounded-below by 0, and for each $\lambda > 0$ the set

$$\left\{ U \in \mathbb{R}^{n \times k} : \|M - UU^T\|_{\ell_1} \leq \lambda \right\}, \tag{14}$$

is compact in $\mathbb{R}^{n \times k}$. Thus, the map $U \to \|M - UU^T\|_{\ell_1}$ is coercive in the sense of (Focardi, 2012, Definition 2.1). Hence, by (Focardi, 2012, Theorem 2.2), the set

$$\operatorname*{argmin}_{U \in \mathbb{R}^{n \times k}} \|M - UU^T\|_{\ell_1}$$

is non-empty. Furthermore, by the Cholesky decomposition (Higham, 2002, Theorem 10.9), for every $L \in P_{k,n}$ there exists some $U \in \mathbb{R}^{n \times k}$ such that $L = UU^\top$. Since, conversely, for every $U \in \mathbb{R}^{n \times k}$ the matrix $UU^\top \in P_{k,n}$ we obtain (i).

Any given $M \in P_n$ is positive semidefinite and therefore $e_1^\top M e_1 \geq 0$, where $e_1 \in \mathbb{R}^n$ has entry 1 in its first component and all other entries equal to 0. Therefore, $M_{1,1} = e_1^\top M e_1 \geq 0$ and in particular, $\sqrt{M_{1,1}} \in \mathbb{R}$. Therefore, the matrix $\tilde{U}$ defined by $\tilde{U}_{i,j} = \sqrt{M_{1,1}} I_{i=j=1}$, where $I_{i=j=1} = 1$ if $1 = i = j$ and 0 otherwise, is in $\mathbb{R}^{n \times 1} \subseteq \mathbb{R}^{n \times k}$. Moreover, $\tilde{U}$ satisfies $\|\tilde{U}\tilde{U}^T\|_{\ell_1} \leq \|M\|_{\ell_1}$. Thus, by the triangle inequality, the set

$$D_M := \left\{ U \in \mathbb{R}^{n \times k} : \|M - UU^T\|_{\ell_1} \leq 2\|M\|_{\ell_1} \right\},$$

is non-empty. Furthermore, by (14) it is compact. In summary,

$$\emptyset \neq \underset{U \in D_M}{\operatorname{argmin}} \|M - UU^T\|_{\ell_1} = \underset{U \in \mathbb{R}^{n \times k}}{\operatorname{argmin}} \|M - UU^T\|_{\ell_1}. \tag{15}$$

Hence $f(M)$, described by condition (ii), is equivalently characterized by

$$f(M) \in \underset{U \in D_M}{\operatorname{argmin}} \|M - UU^T\|_{\ell_1}, \quad \text{for all } M \in P_n. \tag{16}$$

The advantage of (16) over condition (ii) is that the set $D_M$, is compact, whereas $\mathbb{R}^{n \times k}$ is non-compact.

For any set $Z$ denote its power-set by $2^Z$. Define the function $\phi$ by

$$\phi : P_n \to 2^{\mathbb{R}^{n \times k}},$$
$$M \mapsto D_M.$$

Next, we show that $\phi$ is a weakly measurable correspondence in the sense of (Aliprantis & Border, 1999, Definition 18.1). This amounts to showing that for every open subset $\mathcal{U} \subseteq \mathbb{R}^{n \times k}$ the set $\tilde{\mathcal{U}} := \{M \in P_n : \phi(M) \cap \mathcal{U} \neq \emptyset\}$ is a Borel subset of $P_n$.

To this end, define the function

$$G : P_n \times \mathbb{R}^{n \times k} \to \mathbb{R},$$
$$(M, U) \mapsto 2\|M\|_{\ell_1} - \|M - UU^T\|_{\ell_1},$$

and let $p$ be the canonical projection $P_n \times \mathbb{R}^{n \times k} \to P_n$ taking $(M, U)$ to $M$. Observe that, for any non-empty open $\mathcal{U} \subseteq \mathbb{R}^{n \times k}$ we have that

$$\tilde{\mathcal{U}} = p\left[ G^{-1}\left[[0, \infty)\right] \cap (P_n \times \mathcal{U}) \right].$$

Since $G$ is continuous and $[0, \infty)$ is closed in $\mathbb{R}$ then $G^{-1}[[0, \infty)]$ is closed. Since both $\mathbb{R}^{n \times k}$ and $P_n$ are metric sub-spaces of $\mathbb{R}^{n^2}$ then they are locally-compact, Hausdorff spaces, with second-countable topology. Thus (Cohn, 2013, Proposition 7.1.5) implies that the open set $P_n \times \mathcal{U} = \bigcup_{j \in \mathbb{N}} K_j$ where $\{K_j\}_{j \in \mathbb{N}}$ is a collection of compact subsets of $P_n \times \mathbb{R}^{n \times k}$.

Since $P_n$ and $\mathbb{R}^{n \times k}$ are $\sigma$-compact, i.e. the countable union of compact subsets, $P_n \times \mathbb{R}^{n \times k}$ is also $\sigma$-compact by (Willard, 1970, Page 126). Let $\{C_i\}_{i \in \mathbb{N}}$ be a compact cover of $P_n \times \mathbb{R}^{n \times k}$. Since $P_n \times \mathbb{R}^{n \times k}$ is Hausdorff (as both $P_n$ and $\mathbb{R}^{n \times k}$ are), each $C_i \cap G^{-1}[[0, \infty)]$ is compact and therefore $\left\{ K_j \cap \left[ C_i \cap G^{-1}[[0, \infty)] \right] \right\}_{j, i \in \mathbb{N}}$ is a countable cover of $G^{-1}[[0, \infty)] \cap (X \times \mathcal{U})$ by compact sets. Finally, since $p$ is continuous, and continuous functions map compacts to compacts,

$$\begin{aligned}
\tilde{\mathcal{U}} &= p\left[ G^{-1}\left[[0, \infty) \cap (P_n \times \mathcal{U})\right] \right] \\
&= p\left[ \bigcup_{i,j \in \mathbb{N}} \left[ C_i \cap G^{-1}[[0, \infty)] \right] \cap K_j \right] \\
&= \bigcup_{i,j \in \mathbb{N}} p\left[ C_i \cap G^{-1}[[0, \infty)] \cap K_j \right];
\end{aligned}$$

hence $\tilde{\mathcal{U}}$ is an $F_\sigma$ subset of $P_n$ and therefore Borel. In particular, for each open subset $\mathcal{U} \subseteq \mathbb{R}^{n \times k}$, the corresponding set $\tilde{\mathcal{U}}$ is Borel. Therefore, $\phi$ is a weakly-measurable correspondence taking non-empty and compact values in $2^{\mathbb{R}^{n \times k}}$.

Define, the continuous function

$$F : P_n \times \mathbb{R}^{n \times k} \to [0, \infty),$$
$$(M, U) \mapsto \|M - UU^T\|_{\ell_1}.$$

The conditions of the (Aliprantis & Border, 1999, Measurable Maximum Theorem; Theorem 18.19) are met and therefore there exists a Borel measurable function $f$ from $P_n$ to $\mathbb{R}^{n \times k}$ satisfying

$$f(M) \in \underset{U \in D_M}{\operatorname{argmin}} \|M - UU^T\|_{\ell_1} = \underset{U \in \mathbb{R}^{n \times k}}{\operatorname{argmin}} \|M - UU^T\|_{\ell_1},$$

for every $M \in P_n$. This proves (ii).

Fix a Borel probability measure $\mathbb{P}$ on $P_n$. Since $P_n$ is separable and metrizable then by (Klenke, 2013, Theorem 13.6) $\mathbb{P}$ must be a Radon measure. Moreover, since $\mathbb{R}^{n \times k}$ and $P_n$ are locally-compact and second-countable topological spaces, then, the conditions for Lusin's theorem (see (Klenke, 2013, Exercise 13.1.3) for example) are met. Therefore, for every $0 < \varepsilon \leq 1$ there exists a compact subset $K_\varepsilon \subseteq P_n$ satisfying $\mathbb{P}(K_\varepsilon) \geq 1 - \varepsilon$ and for which $f$ is continuous on $K_\varepsilon$. That is, $f|_{K_\varepsilon} \in C(K_\varepsilon, \mathbb{R}^{n \times k})$. Moreover, since $\rho$ is continuous, then

$$f(\cdot)f(\cdot)^\top|_{K_\varepsilon} = \rho \circ f|_{K_\varepsilon} \in \sqrt{C}(K_\varepsilon, P_{k,n}).$$

This gives (iii). $\qquad \square$

*Proof of Theorem 4.3.* Let $\mathcal{N}^{\sigma,\mathrm{narrow}}_{2^{-1}n(n+1),kn}$ denote the collection of deep feed-forward networks in $\mathcal{N}^\sigma_{2^{-1}n(n+1),kn}$ of width at-most $\frac{n(n+2k+1)+4}{2}$. Note that the approximation condition (9) holding for all $\varepsilon > 0$, and all $f \in \sqrt{C}(X, P_{k,n})$ is equivalent to the topological condition $\{\rho \circ \hat{f} \circ \mathrm{vect} : \hat{f} \in \mathcal{N}^{\sigma,\mathrm{narrow}}_{2^{-1}n(n+1),kn}\}$ is dense in $\sqrt{C}(X, P_{k,n})$ for the uniform convergence on compacts topology. We establish the later.

Fix a $\sigma \in C(\mathbb{R})$ satisfying condition 4.2. By (Kidger & Lyons, 2020), $\mathcal{N}^{\sigma,\mathrm{narrow}}_{2^{-1}n(n+1),kn}$ is dense $C(\mathbb{R}^{n(n+1)/2}, \mathbb{R}^{kn})$ in the topology of uniform convergence on compacts.

Let $\phi := h \circ \iota_2 \circ \iota_1$, where $\iota_1 : X \to P_n$, $\iota_2 : P_n \to \mathbb{S}_n$ are the inclusion maps. Since $h$, $\iota_2$, and $\iota_1$ are all continuous and injective, so is $\phi$. Observe that, $g$ is a continuous bijection with continuous inverse. Thus, (Kratsios & Bilokopytov, 2020, Proposition 3.7) implies that $\mathcal{N}^{\sigma,\mathrm{narrow}}_{2^{-1}n(n+1),kn}$ is dense in $C(\phi(X), \mathbb{R}^{kn})$ if and only if $\mathcal{N}^{\sigma,\mathrm{narrow}}_{g,\phi} \triangleq \{g \circ \hat{f} \circ \phi : \hat{f} \in \mathcal{N}^{\sigma,\mathrm{narrow}}_{2^{-1}n(n+1),kn}\}$ is dense in $C(X, \mathbb{R}^{n \times k})$.

Let $R : \mathbb{R}^{n \times k} \ni U \to UU^\top \in P_{k,n}$. Consider the map $R_\star$ sending any $f \in C(X, \mathbb{R}^{n \times k})$ to the map $R \circ f \in \sqrt{C}(X, P_{k,n})$. By (Munkres, 2018, Theorem 46.8) the topology of uniform convergence on compacts on $C(X, \mathbb{R}^{n \times k})$ and $C(X, P_{k,n})$ are equal to their respective compact-open topologies (see (Munkres, 2000, page 285) for the definition) and by (Munkres, 2000, Theorem 46.11) function composition is continuous for the compact-open topology; whence, $R_\star$ is continuous. Moreover, by definition, its image is $\sqrt{C}(X, P_{k,n})$ and therefore, $R_\star$ is a continuous surjection as a map from $C(X, \mathbb{R}^{n \times k})$ to $\sqrt{C}(X, P_{k,n})$. Since continuous maps send dense subsets of their domain to dense subsets of their image, $R_\star\left[\mathcal{N}^{\sigma,\mathrm{narrow}}_{g,\phi}\right] \triangleq \{R \circ g \circ \hat{f} \circ \phi : \hat{f} \in \mathcal{N}^{\sigma,\mathrm{narrow}}_{2^{-1}n(n+1),kn}\} \subset \mathcal{N}^\sigma_{\rho,\phi}$ is dense in $\sqrt{C}(X, P_{k,n})$. As density is transitive, $\mathcal{N}^\sigma_{\rho,\phi}$ is dense in $\sqrt{C}(X, P_{k,n})$. $\quad \square$

*Proof of Corollary 4.4.* By Theorem 4.1 and (8) the map $f^\star : P_n \to P_{k,n}$ is continuous on $K_\varepsilon$. Since $K_\varepsilon$ is compact, Theorem 4.3 implies that there exists some $\hat{f} \in \mathcal{N}^\sigma_{\rho,h}$ of width at-most $\frac{n(n+2k+1)+4}{2}$ satisfying: $\max_{x \in K_\varepsilon} \|f^\star(M) - \hat{f}(M)\|_{\ell_1} < \varepsilon$. $\qquad \square$

### 6.2 Proof of Convergence of Supervised Denise to a Solution Operator of the Learning Problem

*Proof of Theorem 4.6.* By our assumption on $X$ it follows from Corollary 4.4 that for any $\varepsilon > 0$ there exists some $D$ and weights $\tilde{\theta}_D$ such that $\hat{f}_{\tilde{\theta}_D} \in \mathcal{N}^{\sigma,D}_{\rho,h}$ and

$$\max_{M \in X} \left\| f^\star(M) - \hat{f}_{\tilde{\theta}_D}(M) \right\|_{\ell_1} < \varepsilon.$$

Since expectations are taken with respect to $\mathbb{P}$ which is supported on $\mathcal{Z}$ and since the weights $\theta_D$ are chosen to optimize the loss function, we have $\Phi(\theta_D) \leq \Phi(\tilde{\theta}_D)$ and hence

$$\Phi(\theta_D) = \mathbb{E}_{(M,L) \sim \mathbb{P}}\left[\|\hat{f}_{\theta_D}(M) - f^\star(M)\|_{\ell_1}\right]$$
$$\leq \mathbb{E}_{(M,L) \sim \mathbb{P}}\left[\|\hat{f}_{\tilde{\theta}_D}(M) - f^\star(M)\|_{\ell_1}\right]$$
$$\leq \varepsilon.$$

Hence, we can conclude that for any fixed $\varepsilon > 0$, there exists a $D_1 > 0$ such that for all $D > D_1$, we get

$$\mathbb{E}_{(M,L)\sim\mathbb{P}}\left[\|\hat{f}_{\theta_D}(M) - f^\star(M)\|_{\ell_1}\right] \leq \varepsilon\,.$$

In other words, we have that

$$\mathbb{E}_{(M,L)\sim\mathbb{P}}\left[\|\hat{f}_{\theta_D}(M) - f^\star(M)\|_{\ell_1}\right] \xrightarrow{D\to\infty} 0\,,$$

which concludes the proof. $\qquad\square$

### 6.3 Proof of Convergence of the Monte Carlo Approximation

The following Monte Carlo convergence analysis is based on (Lapeyre & Lelong, 2019, Section 4.3). In comparison to them, we do not need the additional assumptions that were essential in (Lapeyre & Lelong, 2019, Section 4.3), i.e. that all minimizing neural network weights generate the same neural network output.

#### 6.3.1 Convergence of Optimization Problems

The following lemma is a consequence of (Ledoux & Talagrand, 1991, Corollary 7.10) and (Rubinstein & Shapiro, 1993, Sec. 2.6, Lemma A1 & Theorem A1 and discussion thereafter).

**Lemma 6.1.** *Let $(\xi_i)_{i\geq 1}$ be a sequence of i.i.d random variables with values in $\mathcal{S}$ and $h : \mathbb{R}^d \times \mathcal{S} \to \mathbb{R}$ be a measurable function. Assume that a.s., the function $\theta \in \mathbb{R}^d \mapsto h(\theta, \xi_1)$ is continuous and for all $C > 0$, $\mathbb{E}(\sup_{|\theta|_2\leq C}|h(\theta,\xi_1)|) < +\infty$. Then, a.s. $f_N : \mathbb{R}^d \to \mathbb{R}, \theta \mapsto \frac{1}{N}\sum_{i=1}^{N}h(\theta,\xi_i)$ converges locally uniformly to the continuous function $f : \mathbb{R}^d \to \mathbb{R}, \theta \mapsto \mathbb{E}(h(\theta,\xi_1))$,*

$$\lim_{N\to\infty}\sup_{|\theta|_2\leq C}\left|\frac{1}{N}\sum_{i=1}^{N}h(\theta,\xi_i) - \mathbb{E}(h(\theta,\xi_1))\right| = 0 \qquad a.s.$$

*Moreover, let the random variables $v_n = \inf_{x\in K}f_n(x)$, consider a minimizing sequence $(x_n)_{n=0}^{\infty}$, given by $f_n(x_n) = \inf_{x\in K}f_n(x)$ and let $v^* = \inf_{x\in K}f(x)$ and $\mathcal{K}^* = \{x \in K : f(x) = v^*\}$. Then $v_n \to v^*$ and $d(x_n,\mathcal{K}^*) \to 0$ a.s.*

#### 6.3.2 Strong Law of Large Numbers

Let $(M_j, L_j)_{j\geq 1}$ be i.i.d. random variables taking values in $\mathcal{Z} = X \times f^\star(X) \subset \mathbb{R}^{n\times n} \times \mathbb{R}^{n\times n} =: \mathcal{S}$. We first remark that $\mathcal{S}$ is a separable Banach space. Moreover, since $f^\star(X)$ is compact as the continuous image of the compact set $X$, it is bounded. Hence, there exists a bounded continuous function $\iota : \mathbb{R}^{n\times n} \to \mathbb{R}^{n\times n}$ such that $\iota|_{f^\star(X)}$ is the identity. Then we define

$$h(\theta, (M_j, L_j)) := \|\iota(L_j) - \hat{f}_\theta(M_j)\|_{\ell_1}$$

where $\hat{f}_\theta \in \mathcal{N}_{\rho,h}^{\sigma,D}$ is a neural network of depth $D$ with the weights $\theta$.

**Lemma 6.2.** *The following properties are satisfied.*

($\mathcal{P}_1$) *There exists $\kappa > 0$ such that for all $Z = (M, L) \in \mathcal{Z}$ and $\theta \in \tilde{\Theta}_D$ we have $\|\hat{f}_\theta(M)\|_{\ell_1} \leq \kappa$.*

($\mathcal{P}_2$) *Almost-surely the random function $\theta \in \tilde{\Theta}_M \mapsto \hat{f}_\theta$ is uniformly continuous.*

*Proof.* By definition of the neural networks with sigmoid activation functions (in particular having bounded outputs), all neural network outputs are bounded in terms of the norm of the network weights, which is assumed to be bounded, not depending on the norm of the input.

Since the activation functions are continuous, also the neural networks are continuous with respect to their weights $\theta$, which implies that also $\theta \in \tilde{\Theta}_M \mapsto \hat{f}_\theta$ is continuous for any fixed input. Since $\tilde{\Theta}_M$ is compact, this automatically yields uniform continuity almost-surely and therefore finishes the proof of ($\mathcal{P}_2$). $\qquad\square$

*Proof of Theorem 4.7.* We apply Lemma 6.1 to the sequence of *i.i.d* random function $h(\theta, (M_j, L_j))$. With $(\mathcal{P}_1)$ of Lemma 6.2 and since $\iota$ is bounded we know that also

$$|h(\theta, (M_j, L_j))| \leq \|\iota(L_j)\|_{\ell_1} + \|\hat{f}_\theta(M_j)\|_{\ell_1}$$

is bounded for $\theta \in \tilde{\Theta}_D$. Hence, there exists some $B > 0$ such that

$$\mathbb{E}_{(M_j, L_j) \sim \mathbb{P}} \left[ \sup_{\theta \in \tilde{\Theta}_D} |h(\theta, (M_j, L_j))| \right] < B < \infty \tag{17}$$

By $(\mathcal{P}_2)$ of Lemma 6.2, the function $\theta \mapsto h(\theta)$ is continuous. Therefore, we can apply Lemma 6.1, yielding that almost-surely for $N \to \infty$ the function

$$\theta \mapsto \frac{1}{N} \sum_{j=1}^{N} h(\theta, (M_j, L_j)) = \hat{\Phi}_s^N(\theta)$$

converges uniformly on $\tilde{\Theta}_M$ to

$$\theta \mapsto \mathbb{E}_{\mathbb{P}}[h(\theta, (M_1, L_1))] = \Phi_s(\theta),$$

where we used that $\iota$ is the identity on $f^\star(X)$.

Let $\Theta_M^{\min} \subset \Theta_D$ be the subset of weights that minimize $\Phi_s$. We deduce from Lemma 6.1 that $d(\theta_{D,N}, \Theta_D^{\min}) \to 0$ a.s. when $N \to \infty$. Then there exists a sequence $(\hat{\theta}_{D,N})_{N \in \mathbb{N}}$ in $\Theta_D^{\min}$ such that $|\theta_{D,N} - \hat{\theta}_{D,N}|_2 \to 0$ a.s. for $N \to \infty$. The uniform continuity of the random functions $\theta \mapsto \hat{f}_\theta$ on $\tilde{\Theta}_D$ implies that $|\hat{f}_{\theta_{D,N}} - \hat{f}_{\hat{\theta}_{D,N}}|_2 \to 0$ a.s. when $N \to \infty$. By continuity of $\iota$ and the $\ell_1$-norm this yields $|h(\theta_{D,N}, (M_1, L_1)) - h(\hat{\theta}_{D,N}, (M_1, L_1))| \to 0$ a.s. as $N \to \infty$. With (17) we can apply dominated convergence which yields

$$\lim_{N \to \infty} \mathbb{E}_{\mathbb{P}} \left[ |h(\theta_{D,N}, (M_1, L_1)) - h(\hat{\theta}_{D,N}, (M_1, L_1))| \right] = 0.$$

Since for every integrable random variable $Z$ we have $0 \leq |\mathbb{E}[Z]| \leq \mathbb{E}[|Z|]$ and since $\hat{\theta}_{D,N} \in \Theta_D^{\min}$ we can deduce

$$\begin{aligned}
\lim_{N \to \infty} \Phi_s(\theta_{D,N}) &= \lim_{N \to \infty} \mathbb{E}_{\mathbb{P}} \left[ h(\theta_{D,N}, (M_1, L_1)) \right] \\
&= \lim_{N \to \infty} \mathbb{E}_{\mathbb{P}} \left[ h(\hat{\theta}_{D,N}, (M_1, L_1)) \right] \\
&= \Phi_s(\theta_D).
\end{aligned} \tag{18}$$

We define $N_0 := 0$ and for every $D \in \mathbb{N}$

$$N_D := \min \left\{ N \in \mathbb{N} \mid N > N_{D-1}, |\Phi_s(\theta_{D,N}) - \Phi_s(\theta_D)| \leq \tfrac{1}{D} \right\},$$

which is possibly due to (18). Then Theorem 4.6 implies that

$$\mathbb{E}_{\mathbb{P}} \left[ \|\hat{f}_{\theta_{D,N_D}}(M) - f^\star(M)\|_{\ell_1} \right] = \Phi_s(\theta_{D,N_D}) \leq \tfrac{1}{D} + \Phi_s(\theta_D) \xrightarrow{D \to \infty} 0,$$

which concludes the proof. $\qquad\square$

### 6.4 Proof of Convergence of Denise in Unsupervised Learning Task

*Proof of Theorem 4.9.* Fix $\epsilon > 0$. Let $f_\epsilon \in \sqrt{C}(\tilde{X}, P_{k,n})$ be such that $\mathbb{E}_{(M,L) \sim \tilde{\mathbb{P}}}[\|M - f_\epsilon(M)\|_{\ell_1}] < \Phi_{\min} + \epsilon$. From Theorem 4.3 we know that there exists some depth $D$ and weights $\tilde{\theta}_D$ such that the resulting neural network $\hat{f}_{\tilde{\theta}_D} \in \mathcal{N}_{g,h}^\sigma$ satisfies $\max_{M \in \tilde{X}} \|f_\epsilon(M) - \hat{f}_{\tilde{\theta}_D}(M)\|_{\ell_1} < \epsilon$. Since $\theta_D \in \Theta_D$ is chosen to minimise $\Phi_u$, we get by triangle inequality

$$\begin{aligned}
\Phi_u(\theta_D) \leq \Phi_u(\tilde{\theta}_D) &= \mathbb{E}_{(M,L) \sim \tilde{\mathbb{P}}}[\|M - \hat{f}_{\tilde{\theta}_D}(M)\|_{\ell_1}] \\
&\leq \mathbb{E}_{(M,L) \sim \tilde{\mathbb{P}}}[\|M - f_\epsilon(M)\|_{\ell_1}] + \mathbb{E}_{(M,L) \sim \tilde{\mathbb{P}}}[\|f_\epsilon(M) - \hat{f}_{\tilde{\theta}_D}(M)\|_{\ell_1}] \\
&\leq \Phi_{\min} + 2\epsilon.
\end{aligned}$$

Since $\Phi_u(\theta_D) \geq \Phi_{\min}$ by definition, we have $|\Phi_u(\theta_D) - \Phi_{\min}| \leq |\Phi_u(\tilde{\theta}_D) - \Phi_{\min}| \leq 2\epsilon$. Using that $\Phi_u(\theta_D)$ is decreasing in $D$, we can conclude that $|\Phi_u(\theta_D) - \Phi_{\min}| \xrightarrow{D \to \infty} 0$. $\qquad\square$

*Proof of Theorem 4.10.* The first two claims follow analogously as in the proof of Theorem 4.7. Choosing the sequence $N_D$ similarly as in the proof of Theorem 4.7 and combining these first two results and Theorem 4.9 via triangle inequality proves that $\hat{\Phi}_u^N(\theta_{D,N_D})$ converges to $\Phi_{\min}$. $\qquad\square$

## 7    Discussion

We provide a simple deep learning based algorithm to decompose positive semidefinite matrices into low rank plus sparse matrices. After the deep neural network was trained, only an evaluation of it is needed to decompose any new unseen matrix. Therefore, the computation time is negligible, which is an undeniable advantage in comparison with the classical algorithms. To support our claim, we provided theoretical guarantees for the recovery of the optimal decomposition. To the best of our knowledge, this is the first time that neural networks are used to learn the low rank plus sparse decomposition for any unseen matrix. The obtained results are very promising. We believe that this subject merits to be further investigation for all online applications where the decomposition must be instantaneous and stable with respect to the inputs.

In future work, Denise's algorithm can be extended to no longer require $M$ to be positive semidefinite such that $M$ can be any $n \times n$ matrix. This can be achieved by replacing the feature map $h$ with a standard vectorization operation. Furthermore, Denise's deep learning architecture can be modified to encode different structures in $L$ by modifying its output layer. E.g. full-rank matrices can be produced using the output layer/readout-map described in (Kratsios & Papon, 2022, Section 3.4.2) and examples of other structures which can be encoded in $L$ by modifying Denise's output layers can be borrowed from Meyer et al. (2011).

### Acknowledgement

We thank Hartmut Maennel, Maximilian Nitzschner, Thorsten Schmidt and Martin Stefanik for valuable remarks and helpful discussions. Moreover, the authors would like to acknowledge support for this project from the Swiss National Science Foundation (SNF grant 179114) and partially funded by the NSERC Discovery grant (RGPIN-2023-04482).

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
