# OpenReview forum: "Denise: Deep Robust Principal Component Analysis for Positive Semidefinite Matrices"
_TMLR — Accepted by TMLR_

### Review · Reviewer_ujHT · 2023-03-08

**Summary Of Contributions:**

This paper proposes Denise, a deep-learning based method for robust principle
components analysis (PCA).
Robust PCA is the problem of decomposing a matrix into the sum of a PSD matrix
and a sparse matrix.
Denise works by using a pre-trained deep neural network to output the Cholesky factor
of the PSD matrix in the robust PCA decomposition given the matrix to decompose
as input.
Pre-training can be supervised using real or synthetic data, or unsupervised
when no robust PCA is available for the training data.
The authors develop several theoretical guarantees for their method:
they show that the robust PCA problem is well-posed --- there exists an almost-everywhere
continuous solution function --- and that this solution function is recovered
as the network becomes infinitely deep.
The paper concludes with experiments on real and synthetic data.

**Audience:**

Yes

**Broader Impact Concerns:**

I have no ethical concerns.

**Claims And Evidence:**

Yes

**Requested Changes:**

- Please incorporate training time for Denise into the time measurements for the experiments so that the comparison is fair.
- Theorem A.1 should either be removed or it's unrealistic assumptions should be carefully discussed in the text.


**Strengths And Weaknesses:**

This is an interesting submission which attempts to circumvent the algorithmic
hardness of robust PCA by pre-training a deep neural network to produce the
decomposition.
This is in contrast to existing works, which use convex relaxations to compute
the decomposition.
The contributions of this work divide naturally into two areas: theoretical and empirical.

On the theory side, the main result is a universal approximation theorem
showing that neural networks can learn to generate (approximate) Cholesky
factorizations of arbitrary input matrices.
This topic is outside my research area and so I cannot comment on the
novelty of these results.
The major downside of the approximation theorems in my opinion is the requirement for exact
optimization of the neural network model, which is NP-Hard in general.
The authors also provide an optimization guarantee for training Denise,
but this relies on a hard-to-justify assumption that the iterates remain in a compact
set.
As a result, I feel this last contribution is negligible.

On the empirical side, the authors show Denise performs comparable to, or better than, existing
convex relaxations on both synthetic and real data.
Although Denise tends to produce denser residuals matrices than PCP, its approximation
error is typically smaller so this may be acceptable to users.
One criticism I have here is that the time to compute the decomposition for
Denise does not incorporate any of the training time for the neural network.
This training time is very significant. For example, the authors report eight
hours of training for the synthetic dataset experiment.
Adjusting for this training time (normalized by the test set) gives that
Denise takes 2880 milliseconds per test example, which is much slower than its
competitors.

It is possible to argue that Dense produces an inference artifact in the sense
of inference compilation [1] and therefore the training time should not be
included in the inference time.
However, the network architecture is specific to the input size of the matrix and the
desired output rank, implying that Denise must be re-trained for each
individual dataset.
As a result, I argue that training time must be incorporated into the comparison
in some fashion for it to be fair.
As is, I do not think the experiments in Tables 1-3 tell the whole story.

To summarize:

**Strengths**:

- The authors provide theoretical guarantees for Denise to recover
    a solution function which computes robust PCAs (e.g. Cholesky factors with sparse residuals).
- Denise performs comparably to or better than competing methods in terms of
    approximation equality.
- Denise allows for very fast inference when the underlying network is pre-trained.


**Weaknesses**:

- Denise is slower than competing methods when adjusted for the time to train
    the underlying neural network.
- The optimization result (Theorem A.1) is not meaningful due to unrealistic assumptions.
- Denise produces less sparse residual matrices than competitors.


### Correctness

I read through all of the proofs. I am confident Theorem 4.1 and Theorem A.1
are correct as stated.
Theorem 4.3 uses theoretical tools I am not familiar with and so I cannot
comment on it's correctness.


### Writing and Presentation:

The paper is well written and the related work is thorough. I would suggest
including references to "inference compilation" methods, since these
work on the same principle of pre-training training a neural network to do a
slow/hard task quickly at test time. A good starting reference is Le, Baydin,
and Wood [1].

There are a few places in the manuscript with awkward work choices (Assumption vs. Lemma)
or where punctuation could be improved.
See "Minor Comments" below.


#### Additional References:

[1] Le, Tuan Anh, Atilim Gunes Baydin, and Frank Wood. "Inference compilation and universal probabilistic programming." Artificial Intelligence and Statistics. PMLR, 2017.

### Questions:

- Theorem 4.6: This assumes (i) you can solve a non-convex optimization problem to global optimum; and (2) the depth of the network can be made arbitrarily large. How can users select the depth in practice, as well as deal with (potentially bad) local minima?
- Section 5.1: "symmetric positive semidefinite sparse matrix $S_0$" --- But, if $S_0$ as symmetric positive-definite, then won't the unsupervised training method be realizable for this dataset? That is, $M = U U^\top$ will hold for some $U$? I thought the only condition on $S_0$ was sparsity for this reason.
- Section 5.1.1: "computed low rank matrix L and the initial low rank matrix $L_0$," --- what does "initial" mean here? Do you mean the low-rank matrix from the synthetic train/test dataset?
- Section 5.1.1: "we approximately obtain a rank of 3 for matrices L" --- By approximately do you mean that the result is not always rank $3$? That is, sometimes the rank is lower or higher? It seems like this would make a large difference for relative error.
- Table 1: What are the numbers in parentheses? Are these standard deviations? If so, what are the taken over? The elements of the test set? Similarly, why is the rank not always an integral number for IALM and PCP? Were the ranks different for some elements of the test set (see question above)
- Tables 1-3: It seems like all of the methods fail at recovering a sufficiently sparse residual matrix $S_0$. That is, $S$ is essentially dense for all the methods except for PCP. Why is that? Is it a thresholding problem? I would have hoped the $\ell_1$ loss would produce more sparsity in the residual.

#### Minor Comments

- Abstract: "current speed optimized method, fast PCP." --- it should be "speed-optimized", since this is a compound adjective.
- Section 1: "In particular, in Finance we need instantaneously" --- there is no need to capitalize finance.
- Section 2: "is an computational improved version" -> computationally
- Section 2: "In particular, The Fast Principal Component Pursuit" --- there is no need to capitalize each word in the initialism.
- Section 2: "In (Herrera et al., 2020), the computation of" --- use "\citet" instead of "\citep".
- Section 2: "Our work is similar to (Gregor & LeCun, 2010) in spirit" --- same as previous.
- Section 2: "with their encoder, as continuity," --- maybe "including" instead of "as"?
- Section 4.4: "It is straight forward to see, that" --- this comma is unnecessary.
- Section 5: "function on a synthetic train dataset"  -> "training"
- Section 5.1.1: "dataset consisting of 10’000 matrices" --- English orthography uses "10,000".
- Assumption 4.4: I think this is meant to be "Corollary 4.4".
- Section 7: "guarantees, for the recovery" --- this comma is unnecessary.
- Appendix A: "it is essential to know that this procedure does converge to a local minimum in our setting" --- convergence to a stationary point (in expectation) as in Theorem A.1 does not guarantee the stationary point is a local minimum.
- Assumption A.2: You typically make assumptions, rather than prove them. Is this supposed to be a Lemma?

---

> ### Author Response · Authors · 2023-03-31
> **Reply to Reviewer ujHT (Part 1)**
>
> Thank you very much for this thorough review. We appreciate your detailed feedback a lot. Below we give point-by-point replies to the raised questions and comments.
>
> **Writing and Presentation:**
> - Thank you for pointing to the reference [1], we now included it in the new Section 5.2 on the computation time of Denise.
>  - You are completely correct that there was something wrong with the Lemma and Corollary commands, as they were shown as Assumptions instead. We corrected this now.
> \end{enumerate}
>
> **Questions:**
> - As with most universal approximation theorems, Theorem 4.6 provides a theoretical guarantee that the proposed deep learning model has the capacity to approximately implement the inverse problem's solution map. That said, in practice, one needs to find a deep enough architecture such that the trained network produces good results empirically, which of course implies that the training itself must have led to a reasonable choice for the network weights. There are some heuristics, however, in practice it is rather done with a trial-and-error approach, where one simply trains the neural network with a stochastic gradient decent variant and optimizes the hyper parameters until the training yields sufficiently good results empirically. Usually, this is relatively easy to achieve, which is the reason why neural networks are so popular nowadays.
>
> There are several directions of research, trying to justify why neural network training works so well empirically.  However, usually these works need simplistic assumptions such that there is no clearly convincing theoretical explanation so far.
>
> Users who want to use Denise can take advantage of our code repository, where our architectural choice and the trained network weights, are made public. This can be used as a starting point when applying Denise to new tasks so that it should be fairly easy to make it work.
>
> - Since we assume that $M$ is symmetric positive semidefinite we need to construct $L$ and $S$ of the synthetic training set such that their sum is symmetric positive semidefinite. The way we chose to do this is by construction both $L$ and $S$ on their own as symmetric positive semidefinite, which implies that their sum is so as well.
>
>     Part of the novelty of our contribution is introducing a connection between proveable deep learning and the sparse recovery problem.  We added details on possible future directions in the discussion section, in particular, variants of Denise which can handle non-symmetric $M$ and variants which can encode full-rank $L$.
>
>     Of course, you are right that $M$ being symmetric positive semidefinite implies that there exists some $U$ s.t.\ $M=UU^\top$, however, this $U$ will in general, not be of the desired rank.
>
> - Yes exactly, we meant the "initial decomposition" given through the data generation; however, we see that this is not entirely clear. We rewrote this now as ``...and the low rank matrix $L_0$ (i.e. the low-rank matrix from the synthetic train and test dataset)''.
>
> - Yes exactly, for PCP and IALM the resulting rank depends on the parameter $\lambda$. We did not optimize this parameter for each of the test samples individually but instead used one choice that yielded good results on average. For some of the matrices in the test set, the rank is lower or higher, as can also be seen from the mean$\pm$std of the rank-metric in Table 1. It is true that for these particular examples this can make a difference in the relative errors. However, when applying these methods to a large number of samples, the only alternative to always achieve rank 3 would be to optimize $\lambda$ for each matrix individually, which would lead to much higher computation times (and might seem to be an unfair comparison from this point of view).
>
> - Yes exactly, in parentheses we show the standard deviations (and the number before is the mean) which are taken over all samples in the test set. We clarified this now in the paper.
>
> - We agree that the sparsity of $S$ is a problem shared by all algorithms, with Denise having the greatest difficulties. We use $\epsilon = 0.01$ as a threshold to count an entry as $0$-entry. Considering the matrices plotted in Figure 1, a threshold to get higher sparsity values, which would better distinguish between the large and the small values, might be something around $0.25$. However, this is arguably far away from $0$, and it is questionable whether one should count values with entries in $(-0.25, 0.25)$ as 0 entries. Therefore, we decided to report the results as we did (as not to make questionable statements); however, we are happy to adjust this if the reviewers think this is justified.
>
>
> **Minor Comments:**
> Thank you again for your very careful reading of our manuscript and for highlighting these typos and mistakes. We corrected all of them as you suggested.

---

> > ### Author Response · Authors · 2023-03-31
> > **Reply to Reviewer ujHT (Part 2)**
> >
> > **Requested Changes:**
> >
> > - We added the new Section 5.2 where we  discuss the computation time of Denise in light of the objections you raised. Of course you are correct that the training time of Denise is the main bottleneck of the method and that we did not discuss this thoroughly enough, which might lead to the impression that the given comparison is unfair. On the other hand, we don't think it is appropriate to distribute the training time to the test samples, since the size of the test set is an arbitrary choice. In particular, using 10K test samples would paint a very bad picture as you outlined correctly, while using 10M test samples (same size as training set) would still lead to outperformance of all competitors.
> >     We hope that the added paragraph puts the given comparison in perspective, by also outlining possible situations where the usage of Denise could be sensible and others where it definitely isn't reasonable.
> >
> > - We agree with your objections against the result in the Appendix and removed it as requested.
> >
> > Sincerely,
> > The Authors

---

> > > ### Comment · Reviewer_ujHT · 2023-03-31
> > > **Thanks for your response**
> > >
> > > Many thanks for responding to my review. Let's continue working together to reach a fair evaluation of this paper.
> > >
> > > > however, this $U$ will in general, not be of the desired rank.
> > >
> > > Right, this makes sense. I didn't notice when writing my review that $M$ is required to be PSD for Denise, but it seems not for other methods of computing robust PCA? It looks like the problem of computing robust PCA is well-defined for general matrices M and there's no mentioned of PSD assumptions until Section 3. Is this correct?
> > >
> > > This also leads me to another question. In general, when does a sparse PCA exist for which $S$ *actually* exhibits a high degree of sparsity? Are there any theoretical guarantees on the sparsity achievable for this problem? Perhaps this is part of the reason most algorithms struggle to produce sparsity in the experiments.
> > >
> > > > We added the new Section 5.2 where we discuss the computation time of Denise.
> > >
> > > Great. I agree with you that including training time in the timing for inference is not fair to your method since it depends on the size of the inference task. Moreover, inference is distinct from training and usually treated separately. However, I do think this information should be provided along with the experimental results to provide a balanced view of the algorithm. What about dividing the time column into "Inference Time"  and "Training Time" and providing both numbers in Tables 1-3? Would this be an acceptable compromise?
> > >
> > > > We agree with your objections against the result in the Appendix and removed it as requested.
> > >
> > > Excellent. I agree with your comments on Review CEtb that understanding gradient-based optimization for neural networks is a unsolved problem; I think it's best to avoid opening up that problem in this work, where the focus is quite different.

---

> > > > ### Author Response · Authors · 2023-04-04
> > > > **Answers to follow-up question.**
> > > >
> > > > Dear Reviewer ujHT,
> > > >
> > > > Thank you very much for you very fast and detailed reply.  The following are our answers:
> > > >
> > > > > Right, this makes sense. I didn't notice when writing my review that is required to be PSD for Denise, but it seems not for other methods of computing robust PCA? It looks like the problem of computing robust PCA is well-defined for general matrices M and there's no mentioned of PSD assumptions until Section 3. Is this correct?
> > > >
> > > >  Yes, it is correct that the robust PCA problem in general does not need the PSD assumption. However, we designed Denise such to hardcode the low rank and PSD structure into its output; the latter of which is needed to ensure meaningful covariance matrix are generated (cf. our real world dataset).  For this we made the assumption that $M$ itself is PSD.
> > > >
> > > > However, as we sketched in above, if the user does not want $M$ to be a covariance matrix then, as long as the low rank part $L$ of its decomposition is PSD, Denise can be modified accordingly.  The necessary ``would-be'' modifications are  outlined in our new discussion Section $5.4$. The theoretical guarantees would need to be adapted, but as we note in the new discussion section, the necessary modifications would be minor to Denise and the proof of its guarantees; in fact less structure would imply simpler proofs and a simpler deep learning model.
> > > >
> > > > >This also leads me to another question. In general, when does a sparse PCA exist for which actually exhibits a high degree of sparsity? Are there any theoretical guarantees on the sparsity achievable for this problem? Perhaps this is part of the reason most algorithms struggle to produce sparsity in the experiments.
> > > >
> > > >  To the best of our knowledge, there is no general result in matrix algebra or in non-convex analysis, which describes broad classes of matrices for which the $\ell^1$ minimization problem guarantees a given degree of sparsity.
> > > >
> > > > That said, e.g. Theorem 1.1 of [1] does give guarantees under which assumptions on $L$ and $S$ PCP should recover the true decomposition. In our training dataset the condition on $S$ to have its non-zero entries uniformly distributed is not satisfied (since we need the symmetry and PSD conditions to be satisfied), hence this result does not apply for our synthetic dataset. Therefore, you are right that this may be part of the reason why the algorithms struggle with the sparsity.  \add{We have added this remark to our numerics section}.
> > > >
> > > > >Great. I agree with you that including training time in the timing for inference is not fair to your method since it depends on the size of the inference task. Moreover, inference is distinct from training and usually treated separately. However, I do think this information should be provided along with the experimental results to provide a balanced view of the algorithm. What about dividing the time column into "Inference Time" and "Training Time" and providing both numbers in Tables 1-3? Would this be an acceptable compromise?
> > > >
> > > > We now included the training time in the tables as you suggested. We only show the training time in those settings in which Denise was actually (re)trained but not in those where it was only evaluated, since we have the feeling that otherwise it could easily be misinterpreted in the sense that Denise had to be trained for each of these evaluation settings separately. We further clarified this in the Sections 5.1 and 5.1.1 of the paper.
> > > >
> > > > >Excellent. I agree with your comments on Review CEtb that understanding gradient-based optimization for neural networks is a unsolved problem; I think it's best to avoid opening up that problem in this work, where the focus is quite different.
> > > >
> > > >
> > > > We fully agree.
> > > >
> > > >
> > > >
> > > > Best regards,
> > > >
> > > > The Authors
> > > >
> > > >
> > > > [1] Emmanuel J. Candes, Xiaodong Li, Yi Ma, and John Wright. Robust principal component analysis? J. ACM, 58(3):11:1–11:37, June 2011. ISSN 0004-5411.

---

### Review · Reviewer_uuQY · 2023-03-14

**Summary Of Contributions:**

This paper introduces a deep learning model to perform robust PCA which has a theoretical guarantee on its accuracy on positive semi-definite matrices. It is based on the universal approximation property of neural networks. It has the advantage to achieve a fast PCA computation on new matrices compared to existing state-of-the-art methods, while maintaining a similar accuracy. On practical data, supervised learning is first used to pre-train the deep learning model on synthetic data, and then unsupervised learning approach is used to fine-tune the model on practical data in order to achieve a good accuracy.



**Audience:**

Yes

**Broader Impact Concerns:**

Overall, I think that there is still a gap between your theoretical results and numerical results. I would be good to improve these aspects in your revision.


**Claims And Evidence:**

Yes

**Requested Changes:**

As mentioned in the introduction: … small perturbations lead to small changes in the output … I am not sure where you have presented these results, could you make this clearer?

Is it possible to include the sparse term S in your assumption 4.5 so that you training data follows L+S?

Regarding the numerical results, I think that you are assuming that you can estimate correctly the rank of the practical dataset in order to apply the supervised training approach, before performing unsupervised fine-tuning (beginning of section 5.1.1). It is not clear how you prepare the synthetic data with the correct rank, could you be more precise on this?

My last comment it to improve the quality of writing. What is the f* in Theorem 4.7? Is it the one in Assumption 4.5, or 4.4? I guess that it is the former. In Theorem 4.3, use Assumption 4.2 rather than Condition 4.2. Typo: train -> training dataset in the beginning of section 5, etc.



**Strengths And Weaknesses:**

In Section 4, this paper presents the main universal approximation result to decompose a psd (positive semi-definite) matrix M into low-rank psd L + sparse matrix S using deep networks. I think this result is very interesting. However, it assumes that the training set follows an exact low-rank model such that the sparse part of the psd matrix S is zero (assumption 4.5). I think this assumption is quite strong, as it guarantees (theorem 4.6,4.7) that the deep learning model works if M = L, but it is not clear what happens if S is non-zero. More discussions are needed.

If I understood, these results are for the supervised training setup. It is still unclear what happens to the unsupervised fine-tuning setup.

---

> ### Author Response · Authors · 2023-03-31
> **Response to Reviewer uuQY**
>
> Thank you very much for your thoughtful review. Below we give point-by-point answers to the raised concerns and explain the changes we made to the manuscript.
>
> We would like to mention that unfortunately, there was a formatting issue that happened when incorporating the TMLR template, which led to Lemmas and Corollaries being displayed as ``Assumptions''. We apologize for any confusion this might have caused. This has since been corrected in the updated version of the paper.
>
> **Strengths and Weaknesses:**
>
>  - We do not assume that the sparse part of the training samples $S=0$, i.e. that $M=L$. We assume that we have a compact subset $X$ of matrices $M$ that are symmetric and PSD and can be of any rank. For this subset we assume that there exists a continuous function $f$ which maps $M$ to the Cholesky-decomposition $U$ of the low-rank part $L$ (of rank at most $k$) of the decomposition of $M$ which minimizes the $\ell_1$-norm of the resulting sparse part $S=M-L$. While it is clear that such as, possibly discontinuous or even non-measurable, function exists which maps every $M \in X$ to such a $U$ (by simply choosing an arbitrary member of the set of minimizers), the technical novelty of our Theorem 4.1 (iii) is that is proves that there is a continuous $f$ on a compact subset of PSD matrices of arbitrarily high probability, with respect to any pre-set probability measure on PSD matrices.  This result is key in applying our universal approximation theorem, since in general, discontinuous functions cannot be uniformly (worst-case) approximated by neural networks with continuous activation functions; a consequence of the uniform limit theorem from topology.
>
> Based on this continuous solution map $f$, solving the inverse-problem, we define the function $f^\star$ which maps $M$ to the low-rank part $L$ of the decomposition (by reversing the Cholesky-decomposition).
>
> Finally, the training set consists of the tuples of $(M,L)$ where $L=f^\star(M)$. We did not include $S = M-L$ (which is in general not 0) in this tuple, because it is uniquely determined by $M$ and $L$.
>
> - You are correct that the convergence result is for the supervised training setup only. The reason for this is that only in the supervised training there is a target function to which the trained algorithm can converge. In the unsupervised setup, no such target function exists, therefore no convergence result of that kind can hold.
>     On the other hand it is clear that training with the unsupervised loss function $\Phi_u$ leads to a resulting decomposition that minimizes the $\ell_1$-norm of the sparse part $S$. We added these results in the new Section 4.5. Thank you for informing us that these results would benefit the paper.
>
>
> **Requested Changes:**
>
> - Since the resulting trained matrix decomposition performed by Denise is continuous with respect to its input $M$ it has the property that small perturbations in the input lead to small changes in Denise's output. The continuity of Denise is a consequence of the use of neural networks, since they are continuous functions by definition.  We note that other methods do not necessarily guarantee this type of stability.
>
>     Furthermore, by Theorem 4.1 (iii), we know that the target function being learned, namely $f^{\star}$, is continuous.  Thus, small perturbations in the training data lead to small perturbations in the learned weights of the deep learning model.  Consequentially, Denise's outputs vary continuously as the training data is perturbed.
>
>     In this way, Denise is robust both to changes in inputs after training, and the input training data itself.
>
>
> - As outlined above, we already consider training data of the form $L+S$ in Assumption 4.5.
>
> - You are right that in order to apply our method we have to fix the rank $k_0$ that we want to use in the supervised and unsupervised training as well as in the evaluation. However, we would argue that the rank of the low-rank matrices $L$ is rather a user choice, i.e. the choice of the user how many principle components he wants to use, than something that has to be inferred from the data (since the data will have full rank $n$ in general). In our case, for simplicity and comparability, we chose to use the rank $k_0=3$ for the real world dataset as we had used it also for the synthetic dataset.
>
>     A different choice of $k_0$ for the real world dataset would mean that also the supervised training needs to be redone with a corresponding new synthetic dataset with the given rank $k_0$.  We added a paragraph in Section 5.3 to clarify this further, thank you for pointing this point of unclarity out.
>
> -You are completely correct that it was not clear which $f^{\star}$ was used in Thm 4.6 and 4.7. We clarified this now.  Additionally, we changed "Condition'' to "Assumption'' in Thm 4.3 and corrected the typo. Thank you for your careful reading.
>
> Sincerely,
> The Authors.

---

### Review · Reviewer_CEtb · 2023-03-28

**Summary Of Contributions:**

This paper proposed a deep learning-based algorithm for robust PCA of covariance matrices which is called Denise. The authors provide some statistical properties of Denise and empirical studies on synthetic and real-world datasets to validate the proposed method.





**Audience:**

Yes

**Broader Impact Concerns:**

I think there is no broader impact concerns since this is a theoretical paper.

**Claims And Evidence:**

Yes

**Requested Changes:**

Please see above.

**Strengths And Weaknesses:**

1. My main consideration is the gap between theoretical analysis and optimization algorithm. The main theoretical Guarantees for Denise are based on the solution of the $\ell_1$ regularized problem, while the training algorithm depends on some smooth approximation. I am not sure whether such approximation still hold the properties of Denise we desired. Especially, the non-smoothness of $\ell_1$-norm plays an important role to obtain the sparsity. It is unclear whether the smoothing in training procedure is reasonable.

2. Appendix A replaces $\varphi_u$ by ${\mathcal C}^2$ function while finding first-order stationary point by SGD only requires the objective function has Lipchitz continuous gradient. I hope the author could explain the benefit from such strong assumption on the approximation function.

3. The real-world dataset used in Section 5.1 contains some 20-by-20 matrices, which is not large. I think applying deep learning model to train such small dataset is not interesting. Some empirical results on large-scale problems is desired.

---

> ### Author Response · Authors · 2023-03-31
> **Response to Reviewer CEtb**
>
> Thank you very much for your review. Below we give point-by-point answers to the raised concerns.
> It is important to mention that unfortunately there was a formatting issue that happened when incorporating the TMLR template, which led to Lemmas and Corollaries being displayed as ``Assumptions''. We apologize for any confusion this might have caused. This has since been corrected in the updated version of the paper.
>
> **Strengths and Weaknesses:**
>
> - You are completely correct that there is a gap between the $\ell_1$ regularized problem that we consider and our convergence analysis of the SGD algorithm (presented in Appendix A), where we used a smoothed version of the $\ell_1$-norm. However, it is important to note that the training algorithm we actually apply in practice as well as all the other theoretical results (in the main part of the paper) do not use the smoothing; i.e., Denise is trained to minimize the $\ell_1$ regularized problem not a smoothed version of it.
>
>     We agree that Theorem A.1 might have led to confusion and decided to remove this result from our paper, in accordance with the request of Reviewer ujHT. The convergence of SGD methods to local or global minima is a large and ongoing research topic on its own.
> We note that results from this field, which provide conditions when our hypotheses are satisfied, could be incorporated with the main results of our present in our work.  In particular, new results providing theoretical convergence guarantees for SGD methods applied to loss functions using the $\ell_1$-norm (or more generally, for loss functions that are not $\mathcal{C}^1$) could readily be applied for Denise.
>     Hence, we agree that it makes more sense not to touch upon this topic in our work.
>
> - You are completely correct that these are very strong assumptions. Your concerns again reinforced us in the decision to remove Theorem A.1 from our work and to focus our theoretical analysis on the approximation and learning capabilities of Denise.
>
> - We agree that $400$ dimensions is not extremely large.  However, dimensionality is only one of the two theoretical bottlenecks to successfully applying any deep learning model, with the other being the target function's regularity.
>     This is confirmed in the optimal approximate rates for ReLU neural networks when approximating continuous functions and smooth functions between Euclidean spaces; see [4] and [3], respectively.
>     Comparable approximation rates can therefore be expected of our Theorem 4.3 if we instead deployed (Theorem 9) [2] *(which is the non-Euclidean analogue of the aforementioned results)* instead of the qualitative non-Euclidean approximation theorem of [1] in its proof.
>     Concretely, approximating an $\alpha$-H\"{o}lder function from a compact subset $X\subset P_{20}$ to $P_{k,20}$ to any given $\varepsilon>0$ precision should a-priori require Denise's deep learning model to have a roughly $\mathcal{O}(\frac1{\varepsilon^{800/\alpha}})$ parameters.
>
>     Since our Theorem 4.1 only guarantees that the solution map $f^{\star}$, which Denise approximates, is continuous on the compact set $K_{\varepsilon}\subset P_{20}$ then it is surprising that any deep learning model can approximate it in practice.  The numerical illustrations show precisely that, even though in this relatively low-dimensional setting, Denise can and does learn to approximate this highly irregular function.
>
>     We agree that the springiness that Denise can approximately solve this learning problem could definitely have been made clearer.  We will accordingly include the above discussion in our paper in a designated ``discussion section'' at the end of the manuscript.
>
> Sincerely,
> The Authors
>
>
> [1] Anastasis Kratsios
> and Ievgen Bilokopytov. Non-euclidean universal approximation. Advances in Neural Information Processing Systems, 2020.
>
> [2] Anastasis Kratsios and Léonie Papon. Universal approximation theorems for differentiable geometric deep
> learning. Journal of Machine Learning Research, 23(196):1–73, 2022. URL
> http://jmlr.org/papers/v23/21-0716.html.
>
> [3] Jianfeng Lu, Zuowei Shen, Haizhao Yang, and Shijun Zhang. Deep network approximation for smooth
> functions. SIAM J. Math. Anal., 53(5):5465–5506, 2021. ISSN 0036-1410. doi: 10.1137/20M134695X. URL https://doi.org/10.1137/20M134695X.
>
> [4] Zuowei Shen, Haizhao Yang, and Shijun Zhang. Optimal approximation rate of ReLU networks in terms
> of width and depth. J. Math. Pures Appl. (9), 157:101–135, 2022. ISSN 0021-7824. doi: 10.1016/j.matpur.2021.07.009. URL https://doi.org/10.1016/j.matpur.2021.07.009.

---

### Decision · Action_Editors · 2023-05-12

**Recommendation:** Accept with minor revision

**Comment:**

Reviewer uuQY mentioned an 1990s paper that may be relevant to the background; if this is appropriate, please include it in a minor revision.

The reviewers had a good discussion, with all four "Leaning Accept" in the end. Reviewer ujHT in particular championed the paper and argued that the novelty and technical contribution were significant; that because TMLR emphasizes "technical correctness over subjective significance", we should not judge the paper by whether the proposed method has immediate impact on practitioners.

After reading the paper myself, I concur with the reviewers. In particular, I thought the range of techniques in the proofs was impressive, and the results were non-trivial.  Hence I'm happy to recommend acceptance of the paper.

**Audience:**

Readers with both neural net and numerical linear algebra background will get the most out of this paper. The theoretical results are in the vein of statistical learning theory, and require a bit of advanced math, at the level of a graduate analysis course. Readers without this analysis background can still understand the main idea (as long as they know some linear algebra).

**Claims And Evidence:**

The paper is in the spirit of some recent scientific work using deep neural nets that circumvent traditional applied math tools (in this case, matrix factorizations) and create neural nets that directly output a quantity-of-interest, in general trading off a large amount of offline training time for very fast inference.  As a larger trend, these ideas are very intriguing, though treated with skepticism as well.

This particular paper proposes a neural net that can factor positive semi-definite matrices into low-rank and sparse components. The paper provides some solid theoretical guarantees, and personally I found the proofs elegant. The theoretical guarantees often have assumptions and shortcomings, but these are standard in the literature (e.g., assuming one can find a global minimizer to a non-convex training problem).

The paper gives numerical experiments as well. The low-rank + sparse factorization is a well-chosen problem since classically it is intractable, and tractable alternatives (those based on convex relaxations) have their own limitations.

I find the numerical experiments the weakest part of the paper, as there are serious limitations (factoring matrices that are just $20\times 20$). However, they serve as a proof-of-concept. Indeed, the third contribution of the paper (besides theory and experiment) is the basic idea itself, and this may have the most value.  These are intriguing ideas, and even if this doesn't change practice overnight, it contributes to a significant change in the field.